# Gauge ambiguities imply Jaynes-Cummings physics remains valid in ultrastrong coupling QED

Adam Stokes[1] & Ahsan Nazir [1]

Ultrastrong-coupling between two-level systems and radiation is important for both fundamental and applied quantum electrodynamics (QED). Such regimes are identified by the breakdown of the rotating-wave approximation, which applied to the quantum Rabi model (QRM) yields the apparently less fundamental Jaynes-Cummings model (JCM). We show that when truncating the material system to two levels, each gauge gives a different description whose predictions vary significantly for ultrastrong-coupling. QRMs are obtained through specific gauge choices, but so too is a JCM without needing the rotating-wave approximation. Analysing a circuit QED setup, we find that this JCM provides more accurate predictions than the QRM for the ground state, and often for the first excited state as well. Thus, Jaynes-Cummings physics is not restricted to light-matter coupling below the ultrastrong limit. Among the many implications is that the system's ground state is not necessarily highly entangled, which is usually considered a hallmark of ultrastrong-coupling.

[1] School of Physics and Astronomy, University of Manchester, Oxford Road, Manchester M13 9PL, UK. Correspondence and requests for materials should be addressed to A.S. (email: adamstokes8@gmail.com) or to A.N. (email: ahsan.nazir@manchester.ac.uk)

Progress in experimental cavity and circuit quantum electrodynamics has granted unprecedented access to the strong, ultrastrong and deep–strong light–matter coupling regimes[1–14]. Recently, circuit QED experiments involving a single $LC$-oscillator mode with frequency $\omega$ coupled to a flux qubit with transition frequency $\omega_m$ have realised coupling $g$ as large as $g/\omega$ ranging from 0.72 to 1.34, with $g/\omega_m \gg 1$[11]. Such regimes offer a new testing ground for the foundations of quantum theory, and offer opportunities for the development of quantum technologies.

Our interest is in material systems that possess anharmonic spectra, and which are commonly truncated to two levels (qubits). In this case, conventional forms of light–matter interaction Hamiltonian yield the so-called quantum Rabi model (QRM) that consists of a linear interaction between the radiation mode and the qubit. Performing the rotating-wave approximation (RWA) then yields the celebrated Jaynes–Cummings model (JCM), which owing to its simple exact solution, has provided deep physical understanding in a wide range of contexts[15–18]. In the ultrastrong-coupling regime where $0.1 < g/\omega < 1$, the RWA is no longer valid[3,5,11] and it is therefore widely believed that the JCM breaks down. For this reason, the QRM is considered indispensible and has found myriad applications in condensed matter, quantum optics and quantum information theory[19–24]. A disadvantage of the QRM when compared with the JCM is the lack of any simple solution, which makes its physical interpretation more difficult[25]. Despite this difficulty, the QRM is known to possess some markedly different physical features compared with the JCM. For example, the JCM predicts that there is no atom–photon entanglement within the ground state, while the ground state of the QRM is highly entangled within the ultrastrong-coupling regime[26].

It was noted some time ago in the context of scattering theory that retaining only a subset of states raises the prospect of gauge non-invariance[27–35]. Yet, when the coupling is weak, it is possible to elicit gauge invariance through systematically accounting for the effects of the truncation[36], and the choice of gauge has no practical implications for the qualitative physical conclusions. It has also been shown in the traditional setting of a single atom weakly coupled to a (multimode) radiation reservoir, that number-conserving (JCM-type) light–matter interaction Hamiltonians can be obtained without recourse to the RWA[37–39].

Very recently, the validity of two-level truncations performed in the Coulomb and multipolar gauges has been assessed[40,41]. The multipolar gauge was found to offer a more accurate QRM than the Coulomb gauge for the particular systems and regimes considered there. This was directly attributed to differences in the corresponding forms of coupling. Specifically, contributions of material levels above the first two were found to be suppressed for dipole-moment matrix elements that feature in the multipolar-gauge coupling, but not for canonical momentum matrix elements that feature in the Coulomb-gauge coupling.

While Refs. [40,41] provide valuable comparisons of the Coulomb and multipolar gauges, we employ a more general approach whereby gauge freedom is encoded into the value of a single real parameter. Our methods are applicable to arbitrary systems in QED, including both cavity and circuit QED implementations. We show that corresponding to a given unique light–matter Hamiltonian, there is a continuous infinity of non-equivalent two-level models, each of which corresponds to a different choice of gauge. We thereby obtain the most general possible Hermitian interaction operator that is bilinear in qubit and oscillator raising and lowering operators, and which is therefore more general than the JCM or QRM forms. We show that a specific choice of gauge, which we call the JC gauge, yields a JCM without any need for the RWA. There are also two gauges that yield distinct QRMs. To understand the implications of our approach within the ultrastrong-coupling regime, we consider in detail a fluxonium-LC-oscillator circuit QED system. We show that the breakdown of the RWA in strong and ultrastrong-coupling regimes does not imply a breakdown of the JCM.

## Results

Our key findings are as follows:

(i) A finite-level truncation of the matter system ruins the gauge invariance of the theory. In the ultrastrong-coupling regime, the predictions relating to the same physical observable are generally significantly different within any two distinct two-level models. However, it remains meaningful to ask which truncation produces the best approximation of the unique physics. We are able to determine the accuracy of approximate two-level models by benchmarking against the unique predictions of the non-truncated (exact and gauge-invariant) theory.

(ii) Each two-level model admits a RWA, which yields a corresponding JCM. The only exception to this occurs in the case of the two-level model associated with the JC gauge, wherein the counter-rotating terms are automatically absent. This JCM is valid far beyond the regime of validity of the RWA as applied to the QRM. It follows that Jaynes–Cummings physics is not necessarily restricted to the weak-coupling regime. In particular, independent of the coupling strength, the ground state is not entangled in the JC-gauge two-level model.

(iii) When focusing on predictions that involve the lowest-lying energy eigenstates of the composite system, the JC-gauge two-level model nearly always outperforms the available QRMs within the regimes of interest. Thus, the JCM can and should be used in various situations previously thought to require use of the QRM.

**Light–matter Hamiltonian.** We first present our approach within the context of cavity QED. We consider a material system with charge $e$ and mass $m$ described by position and velocity variables $\mathbf{r}$ and $\dot{\mathbf{r}}$, respectively, and with potential energy $V(\mathbf{r})$. The material system interacts with an electromagnetic field described by the gauge-invariant transverse vector potential $\mathbf{A}$ and the associated transverse electric field $-\dot{\mathbf{A}} = \mathbf{E}_T$. The total vector potential is given by $\mathbf{A}_{tot} = \mathbf{A} + \mathbf{A}_L$, where the longitudinal part $\mathbf{A}_L$ determines the gauge. In the Coulomb gauge, $\mathbf{A}_L = \mathbf{0}$ so $\mathbf{A}_{tot} = \mathbf{A}$. The scalar potential $A_0$ that then accompanies $\mathbf{A}$ is, upto a factor of $e$, the Coulomb potential. As is well-known, the Maxwell–Lorentz equations are invariant under a gauge transformation taking the form $A_0 \to A_0 - \partial\chi/\partial t$, $\mathbf{A} \to \mathbf{A} + \nabla\chi$, where $\mathbf{A}_L = \nabla\chi$ and $\chi$ is an arbitrary function. Here, we employ a formulation in which this gauge freedom is contained within a single real parameter $\alpha$, which determines the gauge through the function $\chi_\alpha$. This function in turn defines a Lagrangian $L_\alpha$ (see Methods). The value $\alpha = 0$ specifies the Coulomb gauge, while the Poincaré (multipolar) gauge also commonly used in atomic physics is obtained by choosing $\alpha = 1$.

Moving to the Hamiltonian description canonical momenta are defined in the usual way as $\mathbf{p}_\alpha = \partial L_\alpha / \partial \dot{\mathbf{r}}$ and $\mathbf{\Pi}_\alpha = \delta L_\alpha / \delta \dot{\mathbf{A}}$. Quantisation of the system is carried out using Dirac's method[42], full details of which are given in Supplementary Note 1. As in conventional derivations of the QRM and JCM, we restrict our attention to a single-cavity mode. Recently, it was shown that the single-mode approximation can break down in the ultrastrong-coupling regime, and in particular that it eliminates the requisite spatio-temporal structure necessary to elicit causal signal propagation[43]. However, the single-mode approximation does

not result in a breakdown of gauge invariance because gauge transformations remain unitary in the single-mode theory. The generalisation to the multimode case is straightforward[36,39], but is not necessary for understanding the implications of gauge freedom within two-level models. Following conventional derivations, we also make the electric dipole approximation, which similarly does not affect the gauge invariance of the theory.

With these simplifications, the $\alpha$-gauge canonical momenta $\mathbf{p}_\alpha$, $\mathbf{\Pi}_\alpha$ are related to manifestly gauge-invariant observables by

$$m\dot{\mathbf{r}} = \mathbf{p}_\alpha + e(1 - \alpha)\mathbf{A}, \tag{1}$$

$$\mathbf{E}_{\mathrm{T}} = -\mathbf{\Pi}_\alpha - \frac{\alpha\boldsymbol{\varepsilon}(\widehat{\mathbf{d}}\cdot\boldsymbol{\varepsilon})}{\nu}, \tag{2}$$

where $\widehat{\mathbf{d}} = -e\mathbf{r}$ is the material dipole moment, $\nu$ denotes the cavity volume, $\omega$ denotes the cavity frequency and $\boldsymbol{\varepsilon}$ is a cavity unit polarisation vector. The Hamiltonian is the sum of material and cavity energies

$$H = E_{\mathrm{matter}} + E_{\mathrm{cavity}}, \tag{3}$$

where $E_{\mathrm{matter}} = m\dot{\mathbf{r}}^2/2 + V(\mathbf{r})$ and $E_{\mathrm{cavity}} = \nu(\mathbf{E}_{\mathrm{T}}^2 + \omega^2\mathbf{A}^2)/2$. The Hamiltonian is expressible in terms of the $\alpha$-gauge canonical operators using Eqs. (1) and (2), with the well-known Coulomb-gauge ($\alpha = 0$) and Poincaré-gauge ($\alpha = 1$) forms obtained as specific examples.

The energy is a particular example of a gauge-invariant observable, which in Eq. (3) has been expressed as a function of the elementary gauge-invariant observables $\mathbf{x} = \{\mathbf{r}, \dot{\mathbf{r}}, \mathbf{A}, \mathbf{E}_{\mathrm{T}}\}$. More generally, when written in terms of $\mathbf{x}$, any observable $O$ possesses a unique functional form $O \equiv O(\mathbf{x})$. The theory is gauge-invariant in that the predictions concerning any gauge-invariant observable can be calculated using any gauge, and these predictions are unique. The canonical momenta $\{\mathbf{p}_\alpha, \mathbf{\Pi}_\alpha\}$ are, however, manifestly gauge-dependent in that for each different $\alpha$, they constitute different functions of the gauge-invariant observables $\mathbf{x}$. When written in terms of canonical operators $\mathbf{y}_\alpha = \{\mathbf{r}, \mathbf{p}_\alpha, \mathbf{A}, \mathbf{\Pi}_\alpha\}$, an observable $O$ generally possesses an $\alpha$-dependent functional form $O = o^\alpha(\mathbf{y}_\alpha)$. The canonical operators belonging to fixed gauges $\alpha$ and $\alpha'$ are related using the unitary gauge-fixing transformation $R_{\alpha\alpha'} = \exp[\mathrm{i}(\alpha - \alpha')\widehat{\mathbf{d}}\cdot\mathbf{A}]$. This implies that distinct functional forms $o^\alpha$ and $o^{\alpha'}$ of the observable $O$ are related according to

$$O = o^\alpha(\mathbf{y}_\alpha) = R_{\alpha\alpha'}o^\alpha(\mathbf{y}_{\alpha'})R_{\alpha\alpha'}^{-1} \equiv o^{\alpha'}(\mathbf{y}_{\alpha'}). \tag{4}$$

This equation expresses the uniqueness of physical observables independent of the chosen gauge.

The unitarity of the gauge transformation $R_{\alpha\alpha'}$ also ensures that in all gauges, the canonical operators satisfy the canonical commutation relations $[r_i, p_{\alpha,j}] = \mathrm{i}\delta_{ij}$, $[A_i, \Pi_{\alpha,j}] = \mathrm{i}\varepsilon_i\varepsilon_j/\nu$ with all the remaining commutators between canonical operators being zero. These relations allow us to decompose the state space $\mathcal{H}$ of the light–matter system into $\alpha$-dependent matter and cavity state spaces $\mathcal{H}_{\mathrm{m}}^\alpha$ and $\mathcal{H}_{\mathrm{c}}^\alpha$ such that $\mathcal{H} = \mathcal{H}_{\mathrm{m}}^\alpha \otimes \mathcal{H}_{\mathrm{c}}^\alpha$. The eigenstates of the canonical operators $\mathbf{r}, \mathbf{p}_\alpha$ provide a basis for the material space $\mathcal{H}_{\mathrm{m}}^\alpha$ while the eigenstates of the canonical operators $\mathbf{A}, \mathbf{\Pi}_\alpha$ provide a basis for the cavity space $\mathcal{H}_{\mathrm{c}}^\alpha$. It is not possible to define gauge-invariant ($\alpha$-independent) light and matter quantum subsystem state spaces directly in terms of the gauge-invariant observables $\mathbf{x}$, because Eqs. (1) and (2) along with the canonical commutation relations imply that $[m\dot{r}_i, E_{\mathrm{T},j}] = -\mathrm{i}e\,\varepsilon_i\varepsilon_j/\nu \neq 0$.

The present theory yields unique physical predictions despite the $\alpha$-dependence of the quantum subsystems. This is because the representation of an observable by operators is unique, as

expressed by Eq. (4), which implies that the average of an observable $O$ in the state $|\psi\rangle$ is unambiguously $\langle\psi|O|\psi\rangle$. The $\alpha$-dependence of the quantum subsystems is, however, an important feature of the theory, which is made transparent within our formulation. An approximation performed on one of the quantum subsystems will constitute a different approximation in each gauge, and may ruin the gauge invariance of the theory.

**Non-equivalent two-level models.** In conventional approaches, a gauge is chosen at the outset and the Hamiltonian is partitioned into matter and cavity bare energies plus an interaction part. Here, we follow this same procedure, but with the important exception that the gauge is left open rather than fixed. This is achieved through substitution of Eqs. (1) and (2) into Eq. (3), which casts the total Hamiltonian in the form $H = H_{\mathrm{m}}^\alpha(\mathbf{r}, \mathbf{p}_\alpha) \otimes I_{\mathrm{c}}^\alpha + I_{\mathrm{m}}^\alpha \otimes H_{\mathrm{c}}^\alpha(\mathbf{A}, \mathbf{\Pi}_\alpha) + V^\alpha(\mathbf{y}_\alpha)$. Here, $I_{\mathrm{m}}^\alpha$ and $I_{\mathrm{c}}^\alpha$ are the identity operators in $\mathcal{H}_{\mathrm{m}}^\alpha$ and $\mathcal{H}_{\mathrm{c}}^\alpha$, respectively, $H_{\mathrm{m}}^\alpha$ and $H_{\mathrm{c}}^\alpha$ are material and cavity bare energies in $\mathcal{H}_{\mathrm{m}}^\alpha$ and $\mathcal{H}_{\mathrm{c}}^\alpha$, respectively, and $V^\alpha$ denotes the interaction Hamiltonian. The explicit forms of $H_{\mathrm{m}}^\alpha, H_{\mathrm{c}}^\alpha$ and $V^\alpha$ are given in Eqs. (9)–(11) in Methods.

One of the most useful and widespread approximations in light–matter theory is a two-level truncation of the material system, whereby only the first two eigenstates $|\epsilon_0^\alpha\rangle, |\epsilon_1^\alpha\rangle$ of the material bare energy $H_{\mathrm{m}}^\alpha$ are retained. Our approach reveals that this procedure ruins the uniqueness of physical predictions that results from Eq. (4). Using the projection $P^\alpha = |\epsilon_0^\alpha\rangle\langle\epsilon_0^\alpha| + |\epsilon_1^\alpha\rangle\langle\epsilon_1^\alpha|$, we obtain the $\alpha$-gauge two-level model Hamiltonian

$$
\begin{aligned}
H_2^\alpha = \quad & \omega_{\mathrm{m}}\sigma_\alpha^+\sigma_\alpha^- + \omega_\alpha\big(c_\alpha^\dagger c_\alpha + \tfrac{1}{2}\big) + \Delta_\alpha \\
& + \mathrm{i}u_\alpha^-(\sigma_\alpha^+ c_\alpha - \sigma_\alpha^- c_\alpha^\dagger) + \mathrm{i}u_\alpha^+(\sigma_\alpha^+ c_\alpha^\dagger - \sigma_\alpha^- c_\alpha)
\end{aligned}
\tag{5}
$$

where $u_\alpha^\pm = \pm(\mathbf{d}\cdot\boldsymbol{\varepsilon})[\omega_\alpha\alpha \mp \omega_{\mathrm{m}}(1 - \alpha)]/\sqrt{2\omega_\alpha\nu}$ and $\Delta_\alpha = \epsilon_0 + \alpha^2(\mathbf{d}\cdot\boldsymbol{\varepsilon})^2/2\nu$ is an $\alpha$-dependent zero-point shift. The transition dipole moment $\mathbf{d} = \langle\epsilon_1^\alpha| - e\mathbf{r}|\epsilon_0^\alpha\rangle$, which is assumed to be real, is $\alpha$-independent, because $\mathbf{r}$ commutes with $R_{\alpha\alpha'}$. The material Hamiltonian's eigenvalues $\epsilon_0$ and $\epsilon_1 = \omega_{\mathrm{m}} + \epsilon_0$ corresponding to material states $|\epsilon_0^\alpha\rangle$ and $|\epsilon_1^\alpha\rangle$, respectively, are also $\alpha$-independent because $H_{\mathrm{m}}^\alpha = R_{\alpha\alpha'}H_{\mathrm{m}}^{\alpha'}R_{\alpha\alpha'}^{-1}$. The complete derivation of Eq. (5) is given in Methods.

An important topic relating to two-level models and the choice of gauge in light–matter physics concerns the occurrence or otherwise of a superradiant phase transition in the Dicke model at strong coupling[40,44–48]. A precursor already occurs in the QRM, whereby beyond a critical coupling point, an exponential closure of the first transition energy occurs[49–51]. We note that in Eq. (5), counter-rotating and number-conserving interactions generally have different coupling strengths, and a strict bound cannot be given for either coupling independent of the material potential, except if $\alpha = 0$. It follows that the standard "no-go theorem" concerning the ground-state instability of a single dipole, holds in general only in the Coulomb gauge[41,44–47]. An arbitrary-gauge analysis of this topic is important, but lies beyond the scope of this article and will be discussed elsewhere.

We are concerned with the $\alpha$-dependence of predictions obtained when using the Hamiltonian in Eq. (5). This Hamiltonian has neither JC nor Rabi form, because $|u_\alpha^+| \neq |u_\alpha^-|$ and $u_\alpha^+ \neq 0$ except when particular values of $\alpha$ are chosen. Specifically, two distinct QRMs are obtained for the choices $\alpha = 0$ and $\alpha = 1$, which are nothing but the Coulomb and Poincaré-gauge QRMs frequently encountered in quantum optics. On the other hand, by choosing $\alpha = \alpha_{\mathrm{JC}}$, which solves the coupled equations $\alpha_{\mathrm{JC}}(\omega_{\mathrm{m}} + \omega_{\mathrm{JC}}) = \omega_{\mathrm{m}}$ and $\omega_{\mathrm{JC}}^2 = \omega^2 + e^2(1 - \alpha_{\mathrm{JC}})^2/m\nu$, we obtain $u_{\mathrm{JC}}^+ \equiv 0$ and $u_{\mathrm{JC}}^- = -2(\mathbf{d}\cdot\boldsymbol{\varepsilon})\omega_{\mathrm{m}}\sqrt{\omega_{\mathrm{JC}}}/[\sqrt{2\nu}(\omega_{\mathrm{JC}} + \omega_{\mathrm{m}})]$. This choice

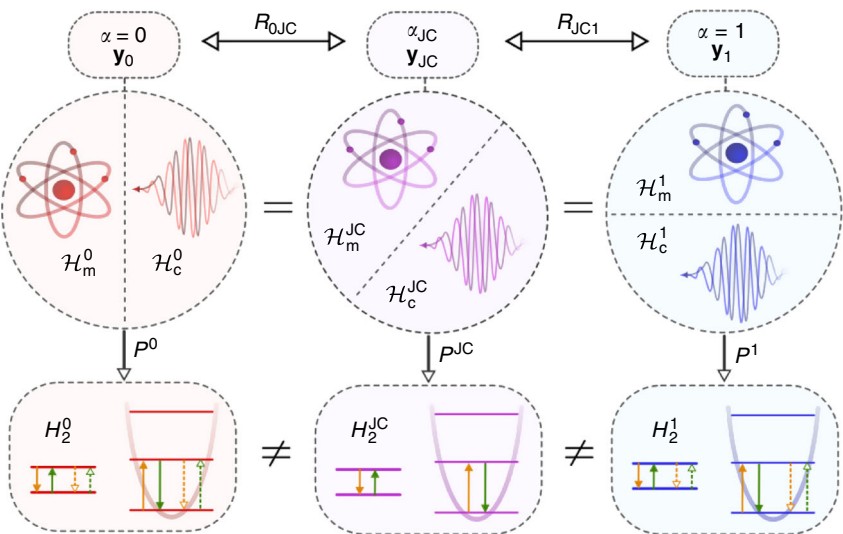

**Fig. 1** Three important gauges and their non-equivalent two-level models. Shown are the $\alpha = 0$, $\alpha = \alpha_{JC}$ and $\alpha = 1$ gauges, and their associated two-level truncations. The different gauges are associated with different unitarily related canonical operators $\mathbf{y}_0$, $\mathbf{y}_{JC}$ and $\mathbf{y}_1$, respectively, which induce different subsystem decompositions of the light–matter Hilbert space. The composite Hilbert space and the Hamiltonian are unique, but a projection onto the first two levels of the material system results in distinct two-level models with Hamiltonians $H_2^0$, $H_2^{JC}$ and $H_2^1$ respectively. The $\alpha = 0$ and $\alpha = 1$ gauge two-level model interaction Hamiltonians both have Rabi form and therefore describe real processes represented by the solid green and orange arrows, as well as counter-rotating processes represented by the dashed arrows. The $\alpha_{JC}$-gauge two-level model interaction has Jaynes–Cummings form and therefore all processes it describes are real

therefore yields a JC Hamiltonian without any need for the RWA. The JCM derived in this way possesses the same advantage of exact solvability as conventional JCMs obtained as RWAs of the Coulomb and Poincaré-gauge QRMs. However, the states $|\epsilon_{JC}\rangle$, operators $\sigma_{JC}^{\pm}$ and parameters $u_{JC}^{-}$, $\omega_{JC}$ are different from their counterparts within conventional JCMs. In particular, the renormalised cavity frequency $\omega_{JC}$ together with the zero-point shift $\Delta_{JC}$ yields a ground-state energy that is a non-constant function of the Coulomb-gauge and multipolar-gauge QRM coupling parameters.

Having derived an expression for the energy, most properties of practical interest can now be calculated using the two-level model associated with any gauge. This includes atomic populations and coherences, as well as various cavity properties, such as photon number. It is, however, possible to go further by defining the two-level representation of any additional observable of interest $O$ as $O_2^{\alpha} = P^{\alpha} O P^{\alpha}$. Restricting the state space $\mathcal{H}_m^{\alpha}$ to the two-dimensional subspace spanned by the eigenstates $|\epsilon_0^{\alpha}\rangle$, $|\epsilon_1^{\alpha}\rangle$ then completes the construction of the two-level model.

Two-level models corresponding to distinct gauges $\alpha$ and $\alpha'$ must be distinguished, because when $\alpha \neq \alpha'$, the projection $P^{\alpha}$ involves all eigenstates of $H_m^{\alpha'}$, and similarly $P^{\alpha'}$ involves all eigenstates of $H_m^{\alpha}$. This is because the gauge transformation does not have product form; $R_{\alpha\alpha'} \neq R_m \otimes R_c$. A pictorial representation of the relationship between different gauges and their associated two-level models is given in Fig. 1. After a two-level truncation, the uniqueness of the representation of observables expressed by Eq. (4) no longer holds, that is, $O_2^{\alpha} \neq O_2^{\alpha'}$ when $\alpha \neq \alpha'$. Distinct two-level models will therefore give different predictions for the same physical quantity. An observable of particular importance is the energy represented by the Hamiltonian, which we focus on hereafter. There is generally no simple relation between distinct two-level model Hamiltonians $H_2^{\alpha}$ and $H_2^{\alpha'}$, when $\alpha \neq \alpha'$. In fact, it was noted some time ago that two-level models associated with different gauges can give different results even in the weak-coupling regime[52]. However, provided that the two-level modification of the operator algebra is accounted for, it can be

shown that certain two-level model predictions are gauge-invariant upto order $d^2$ [36]. This is discussed in more detail in Supplementary Note 2. Regardless, one expects predictions of two-level models corresponding to different gauges to be significantly different when the coupling is sufficiently strong. We show how a comparison of the predictions of different two-level models can be achieved for an arbitrary observable in Methods. We show further that if the material system is a harmonic oscillator, then it is possible to derive a JCM that is necessarily more accurate than any derivable QRM for finding ground-state averages.

**Application to ultrastrong coupling in circuit QED.** When considering less artificial systems than a material oscillator, the relative accuracies of two-level models is more difficult to determine. We now consider an experimentally relevant circuit QED setup consisting of a fluxonium atom coupled to an $LC$ oscillator. The fluxonium is described by the flux variables $\phi$, $\dot{\phi}$ and the external flux $\phi_{\text{ext}}$, along with three energy parameters $E_c$, $E_J$ and $E_l$ that are the capacitive energy, tunnelling Josephson energy and inductive energy, respectively. The external flux $\phi_{\text{ext}} = \pi/2e$ specifies maximum frustration of the atom. The $LC$ oscillator is described by analogous flux variables $\theta$, $\dot{\theta}$, with inductance $L$ and capacitance $C$ defining the oscillator frequency $\omega = 1/\sqrt{LC}$.

In terms of $\mathbf{x} = \{\phi, \theta, \dot{\phi}, \dot{\theta}\}$, the functional form of an observable $O$ is unique $O \equiv O(\mathbf{x})$. On the other hand, different canonical operators $\mathbf{y}_{\alpha} = \{\phi, \xi_{\alpha}, \theta_{\alpha}, \zeta\}$ are related by $\theta_{\alpha} = R_{0\alpha}^{-1} \theta_0 R_{0\alpha}$ and $\xi_{\alpha} = R_{0\alpha}^{-1} \xi_0 R_{0\alpha}$, where $R_{0\alpha} = e^{i\alpha\zeta\phi}$ is a unitary gauge transformation with $\alpha$ real and dimensionless. Here, $\xi_{\alpha}$ and $\zeta$ are canonical momenta conjugate to $\phi$ and $\theta_{\alpha}$, respectively. The gauge choices $\alpha = 0$ and $\alpha = 1$ are called the charge gauge and flux gauge, respectively[53]. The Hamiltonian $H$ describing the system is derived in Supplementary Note 3 and is given in Methods.

In exactly the same way as for the cavity QED Hamiltonian, the projection $P_{\alpha}$ onto the first two eigenstates $|\epsilon_0^{\alpha}\rangle$, $|\epsilon_1^{\alpha}\rangle$ of the material bare energy $H_m^{\alpha}$ can be used to obtain an $\alpha$-dependent

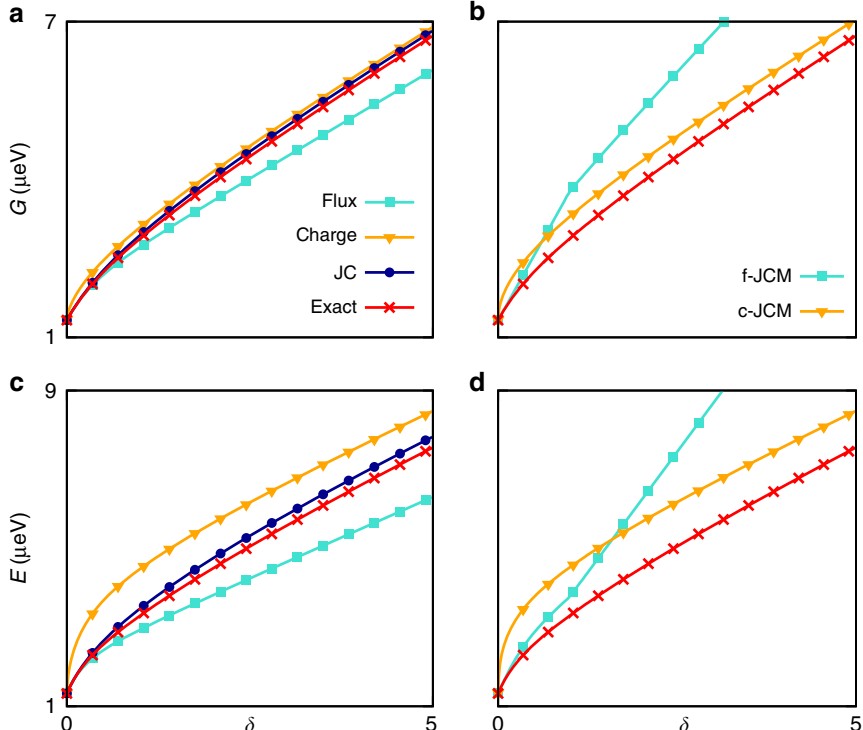

**Fig. 2** Lowest energy levels as functions of detuning. **a** $E_l = 0.33$ μeV, $E_J = 10E_l = E_c$, $\phi_{ext} = \pi/2e$ and $\eta = 1$. The ground energy is plotted with $\delta$ for the flux-gauge and charge-gauge QRMs, for the JC-gauge two-level model and for the exact model. **b** Same as (**a**) for the charge and flux-gauge JCMs obtained as RWAs of the corresponding QRMs. **c** Same as (**a**) for the first excited energy. **d** Same as (**c**) for the charge and flux-gauge JCMs

two-level model Hamiltonian, which at maximal frustration reads

$$
\begin{aligned}
H_2^\alpha = {} & \omega_m \sigma_\alpha^+ \sigma_\alpha^- + \omega_\alpha \left( c_\alpha^\dagger c_\alpha + \tfrac{1}{2} \right) + \Delta_\alpha \\
& + u_\alpha^- \left( \sigma_\alpha^+ c_\alpha + \sigma_\alpha^- c_\alpha^\dagger \right) + u_\alpha^+ \left( \sigma_\alpha^+ c_\alpha^\dagger + \sigma_\alpha^- c_\alpha \right),
\end{aligned}
\tag{6}
$$

where $u_\alpha^\pm = \varphi[\alpha \omega_\alpha \mp (1-\alpha)\omega_m]/\sqrt{2\omega_\alpha L}$ and $\Delta_\alpha = \epsilon_0 + \alpha^2 \varphi^2 / 2L$, in which $\varphi = \langle \epsilon_1^\alpha | \phi | \epsilon_0^\alpha \rangle = \varphi^*$ and $\epsilon_0$ denotes the ground energy of $H_m^\alpha$. The two-level system parameters $\omega_m$, $\varphi$ and $\epsilon_0$ depend implicitly on $E_c$, $E_J$, $E_l$ and $\phi_{ext}$. The renormalised cavity frequency is $\omega_\alpha = \omega \sqrt{1 + 2E_c(1-\alpha)^2 C/e^2}$. Away from the maximal frustration point, the flux $\phi$ possesses diagonal matrix elements in the basis $\{|\epsilon_0^\alpha\rangle, |\epsilon_1^\alpha\rangle\}$, such that $\sigma_\alpha^+ \sigma_\alpha^-$ and $\sigma_\alpha^- \sigma_\alpha^+$ are also linearly coupled to the mode operators $c_\alpha, c_\alpha^\dagger$. In analogy to the cavity QED case, the charge and flux gauges yield distinct Rabi Hamiltonians, but there also exists a value $\alpha = \alpha_{JC} = \omega_m/(\omega_m + \omega_{JC})$ such that $u_\alpha^+ \equiv 0$, which casts the Hamiltonian in JC form.

The ratio $\delta = \omega/\omega_m$ in which $\omega_m$ is taken as the qubit transition at maximal frustration $\phi_{ext} = \pi/2e$, specifies the relative qubit–oscillator detuning. To quantify the relative coupling strength, we use the ratio $\eta = g/\omega$ where $g = \varphi\sqrt{\omega/2L}$. The parameters $g$ and $\omega$ are the coupling strength and cavity frequency of the flux-gauge QRM, but we note that the corresponding parameters associated with any other two-level model could also be used. For different $\alpha$, the $\alpha$-dependent two-level truncation yields different predicted behaviour of physical observables as functions of the model parameters $\delta$, $\eta$ and $\phi_{ext}$. In contrast, the exact predictions resulting from the non-truncated model are $\alpha$-independent (gauge-invariant).

We begin by determining how the ground energy $G$ and first excited energy $E$ vary with the detuning $\delta$ at maximal frustration $\phi_{ext} = \pi/2e$ and fixed coupling $\eta = 1$ (Fig. 2). Regimes with large $\delta$ are presently more experimentally relevant[11,13,14], yet, unless $\delta$ is relatively small ($\delta < 1$), we find that all two-level models become inaccurate in

predicting eigenvalues $E_n > E$ of the non-truncated Hamiltonian. This can be traced to the occurence of resonances in energy shifts, which occur for large $\delta$ (see Supplementary Note 4). Indeed, deviations from the predictions of the QRM have been observed experimentally for such $E_n$ within the ultrastrong-coupling regime[14].

We focus primarily on the experimentally relevant large $\delta$ regime by choosing $\delta = 5$. Other detunings may also be considered and various results for the cases $\delta = 1$ (resonance) and $\delta = 1/5$ are presented in Supplementary Note 5. In Fig. 3a, b, we compare the ground and first excited energies found using various two-level models with the corresponding gauge-invariant energies of the exact theory. The ground and excited-level shifts are obtained by subtracting the corresponding (bare) eigenenergies of the non-interacting system. At maximal frustration, the shift of the ground state can be identified as the Bloch–Siegert shift[4]. The first transition shift is the difference between the ground and excited shifts, and is commonly termed the Lamb shift by analogy with atomic hydrogen[14]. In the RWA, the coupling-dependent zero-point contribution $\omega_\alpha/2 + \Delta_\alpha$ in Eq. (6) gives the ground energy. For $\alpha \neq \alpha_{JC}$, this results in an incorrect expression for the Lamb shift even for weak coupling[36,54] (see also Supplementary Note 2). It is therefore unsurprising that the flux and charge-gauge JCMs are inaccurate in predicting the associated dressed energies within the ultrastrong-coupling regime, as illustrated in Fig. 3a, b. In contrast, for the two-level model of the JC gauge ($\alpha = \alpha_{JC}$), the RWA is no longer an approximation. The ground energy $\omega_{JC}/2 + \Delta_{JC}$, is different from the results of the RWA applied in the $\alpha = 0$ and $\alpha = 1$ gauges, and it does lead to the expected expression for the Lamb shift within the weak-coupling regime[36] (see Supplementary Note 2). Thus, even though the Hamiltonian has Jaynes–Cummings form, it is not evident that like the charge and flux-gauge JCMs, the JC-gauge two-level model will necessarily be inaccurate in predicting dressed energies within the ultrastrong-coupling regime. Indeed,

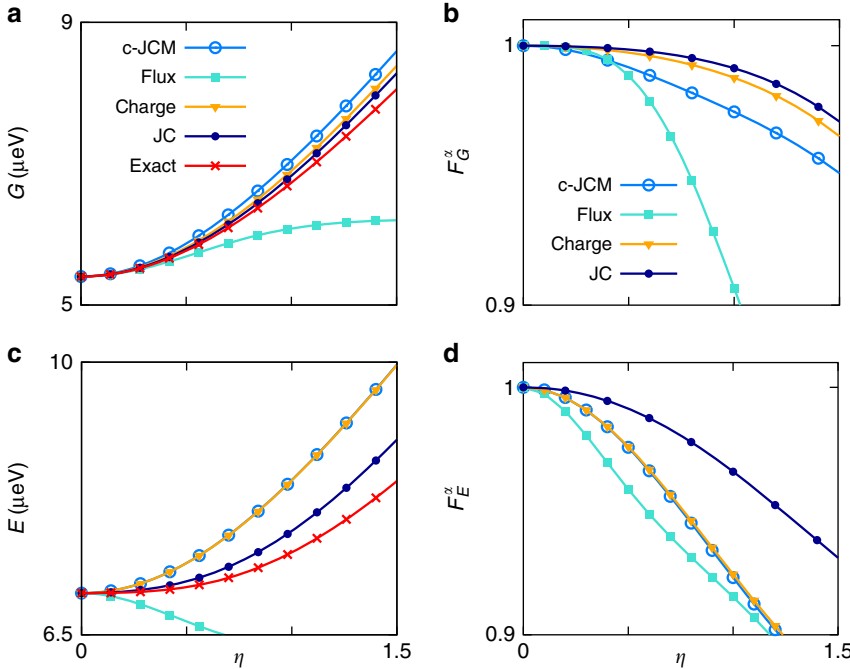

**Fig. 3** Lowest energies and eigenstate fidelities as functions of coupling strength. In all plots, $E_l = 0.33\,\mu\text{eV}$, $E_J = 10E_l = E_c$, $\delta = 5$ and $\phi_{\text{ext}} = \pi/2e$. **a** The ground energy is plotted with $\eta$ for the flux-gauge and charge-gauge QRMs, for the JC-gauge two-level model, for the exact model and for the charge-gauge JCM obtained via the RWA. The flux-gauge JCM is extremely inaccurate in this regime and is not shown. **b** The ground-state fidelity $F_G^\alpha$ is plotted with $\eta$ for the flux-gauge $\alpha = 1$ and charge-gauge $\alpha = 0$ QRMs, for the JC gauge and for the charge-gauge JCM (c-JCM) that is obtained as the RWA of the corresponding Rabi model. **c** Same as (**a**) for the first excited energy. **d** Same as (**b**) for the first excited state. For the excited state, the RWA remains valid in the charge gauge

Fig. 3a, b shows that the JC-gauge two-level model is not only more accurate than the flux and charge-gauge JCMs, it is also more accurate than the flux and charge-gauge QRMs.

To determine which two-level model yields the most accurate lowest energy eigenstates, we compute the ground- and first excited-state fidelities $F_G^\alpha = \left|\langle G_2^\alpha|G\rangle\right|^2$ and $F_E^\alpha = \left|\langle E_2^\alpha|E\rangle\right|^2$, where $|G\rangle$ and $|E\rangle$ are the exact ground and first excited eigenstates of the non-truncated Hamiltonian $H$, while $\left|G_2^\alpha\right\rangle$ and $\left|E_2^\alpha\right\rangle$ are the corresponding eigenstates of $H_2^\alpha$. Figure 3c, d shows that the JC-gauge model is more accurate than both QRMs, and much more accurate than conventional JCMs, especially in the case of the ground state. Since the JC-gauge two-level model tends to produce a more accurate representation of the lowest two energy states of the system, it is natural to suppose that it will generally be more accurate than the QRM in predicting observable averages in these states. This is verified for the cases of ground-state photon number averages in Supplementary Note 6.

To link with recent experiments in which circuit properties are measured for varying external flux $\phi_{\text{ext}}$, Fig. 4 shows the behaviour with $\phi_{\text{ext}}$ of the lowest dressed energies when $\eta = 1/2$. The JC-gauge again yields the most accurate two-level model (Fig. 4a, b) despite the clear breakdown of the RWA (Fig. 4c, d). It follows that Jaynes–Cummings physics is not synonymous with the RWA, and that a departure from Jaynes–Cummings physics is not implied within the ultrastrong-coupling regime. For larger $\eta$, two-level models become increasingly inaccurate, though the JC gauge continues to give the best agreement with exact energies even within the deep–strong coupling regime (see Supplementary Note 5).

## Discussion

The behaviour shown in Figs. 2–4 can be understood by deriving an effective Hamiltonian valid sufficiently far from resonance

(dispersive regime)[55], details of which are given in Supplementary Note 4. In this context, let us first consider the flux gauge, wherein the light and matter systems are coupled through the material position operator $\phi$. The matrix elements of this operator between material states $|\epsilon_n^1\rangle$ are largest between adjacent levels $n$, $n \pm 1$[41] (see Supplementary Note 4). Thus, provided higher material levels are sufficiently separated from the lowest two, the coupling to them can be neglected, unless the light–matter coupling $\eta$ is very large, or $\delta$ is large enough that several material energies lie within the first oscillator band $\omega$. For such large $\delta$, contributions of energy denominators in the effective Hamiltonian become large due to the occurence of resonances $\epsilon_{ni} \sim \omega$, $\epsilon_{ni} = \epsilon_n - \epsilon_i$, $i = 0, 1, n > 1$ (see Supplementary Note 4). The flux-gauge QRM is therefore qualitatively accurate if $\delta$ and $\eta$ are sufficiently small. This includes accurately predicting higher system energy levels $E_n > E$ as well as the first two levels $G$ and $E$[41] (see Supplementary Note 5).

In the charge gauge, the light–matter coupling occurs via the material canonical momentum $\xi_0$, for which matrix elements involving higher levels are not suppressed (see Supplementary Note 4). Independent of $\delta$, when the coupling is sufficiently large, they cannot generally be neglected even for highly anharmonic material spectra, so the charge-gauge QRM generally breaks down[41]. However, the ratio of the flux-gauge QRM-coupling strength $g$ and the coupling strength $\tilde{g}_0$ of the charge-gauge QRM, increases as $\delta$ increases (see Supplementary Note 4). For large enough $\delta$, the charge-gauge coupling is significantly weaker than that of the flux gauge to the extent that for sufficiently large $\delta$ and provided $\eta$ does not become too large, the charge-gauge QRM is qualitatively accurate for the ground level $G$, and occasionally for the first level $E$ (Figs. 2–4).

In the general $\alpha$-gauge, all flux-gauge coupling terms are weighted by $\alpha$ and all charge-gauge coupling terms by $1 - \alpha$. By tuning $\alpha$, the $\alpha$-gauge two-level model smoothly interpolates

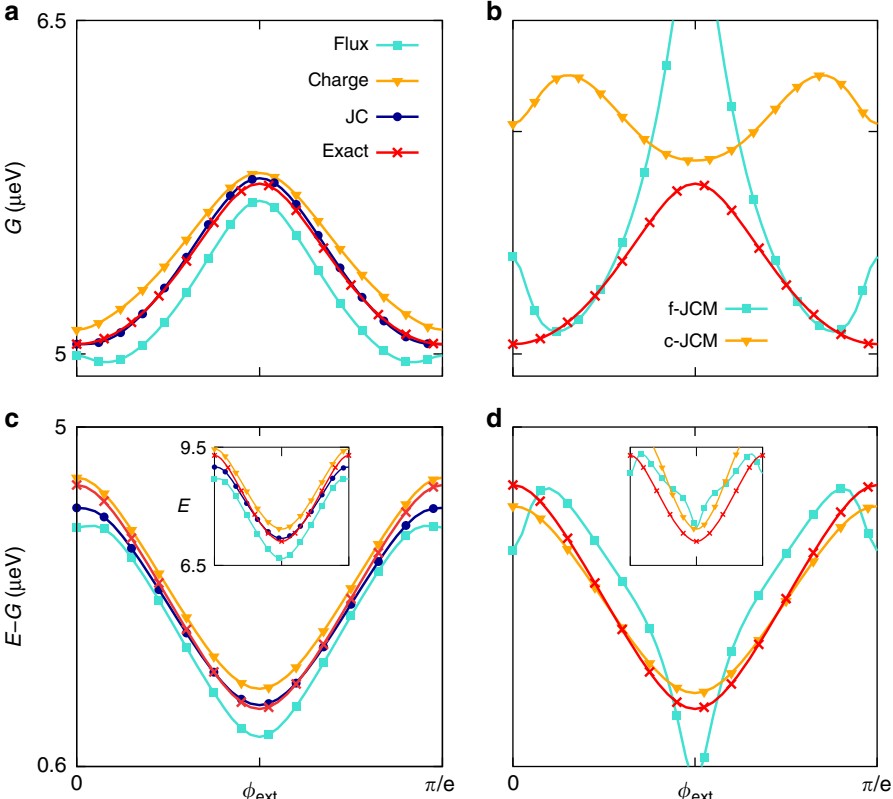

**Fig. 4** Lowest energies as functions of external flux. In all plots, $E_l = 0.33$ μeV, $E_J = 10E_l = E_c$, $\delta = 5$ and $\eta = 0.5$. **a** The ground energy $G$ is plotted with $\phi_{ext}$ for the flux-gauge and charge-gauge QRMs, for the JC-gauge two-level model and for the exact model. **b** For the same range as (**a**) the ground energy is plotted for the flux-gauge and charge-gauge JCMs that are obtained as RWAs of the corresponding Rabi models, and for the exact model. **c** Same as (**a**) for the first transition energy $E - G$. The inset shows the corresponding first excited energy $E$. The JC gauge is generally the most accurate two-level model, but because the charge-gauge QRM overestimates both the ground and excited energy, it becomes relatively accurate in the transition $E - G$ for fluxes away from the maximal frustration point. **d** Same as (**b**) for the first transition energy. The inset shows the corresponding excited energies over the same range as the inset in (**c**). We see that especially in the case of the charge gauge, the JCM is inaccurate for the ground and excited energies, but it is by comparison more accurate for the transition

between the two available QRMs. In particular, the $\alpha_{JC}$-gauge JCM is defined such that the counter-rotating terms that give the dominant contribution to deviations between the exact and two-level model ground states are eliminated (see Supplementary Note 4). This allows us to understand why the $\alpha_{JC}$-gauge JCM accurately represents the ground state across all parameter regimes. As $\delta$ and $\eta$ increase, the $\alpha_{JC}$ gauge becomes predominantly charge-like (see Supplementary Note 4) and like the charge-gauge QRM becomes inaccurate for predicting levels $E_n > E$.

Quite generally, two-level models remain most accurate in predicting the first two system levels $G$ and $E$. For the lowest such levels of certain circuit QED systems, spectroscopic experimental data have been matched to the predictions of the QRM defined by the Hamiltonian $h = -(\Delta\sigma^z + \epsilon\sigma^x)/2 + \omega a^\dagger a + g'\sigma^x(a + a^\dagger)$, where $\Delta$ and $\epsilon$ are tunnelling and bias parameters, respectively, and $g'$ denotes the coupling strength[11,13,14]. In Ref. [13] for example, the parameters $\Delta$, $g'$ and $\omega$ are treated as constant fitting parameters, while $\epsilon$ is externally variable. It is important to note, however, that fitting transitions between eigenenergies of $h$ to experimental data does not preclude the possibility of fitting other models to experimental data.

It is possible to rotate the flux-gauge QRM $H_2^1$ into the form of $h$, but upon doing so, each of $\Delta$, $\epsilon$ and $g'$ are found to be non-trivial functions of $\phi_{ext}$. In particular, for the fluxonium-$LC$ system we consider, $g'$ and $\Delta$ do not remain constant while varying $\epsilon$ by varying $\phi_{ext}$. Moreover, the $\alpha$-gauge two-level model cannot be uniquely specified in terms of the parameters of $h$. Whenever

$\phi_{ext} \neq \pi/2e$, these properties obstruct meaningful comparison between our results and experimental results of the kind found for example in Ref. [13].

More relevant experimental results for the system we consider are given in Ref. [2], where spectroscopic data were found to agree well with the non-truncated fluxonium-$LC$ Hamiltonian $H$ of Eq. (13). There, the fluxonium energies $E_c$, $E_l$ and $E_J$ were treated as fitting parameters. Our results show that using such a fitting procedure, the JC-gauge two-level model would offer better agreement with experimental data than the QRM, at least for the lowest two levels $G$ and $E$. This occurs over the full range of $\delta$ shown in Fig. 2 with only a few exceptions in the case of the excited state $E$ when $\delta$ is small (see Supplementary Note 5).

The results presented here open up multiple avenues for further investigation. For example, our more general form of two-level model in which the gauge is left open is capable (albeit fortuitously) of exactly predicting a given energy value, but it remains to be understood in more detail. A comprehensive comparison of different methods for deriving two-level model descriptions is also yet to be performed.

An investigation of the implications of the arbitrary-gauge formalism for the occurence of phase transitions in multi-dipole systems constitutes further important work. The dependence on arbitrary-gauge parameters of weaker truncations such as three-level atomic models remains to be investigated as does the generalisation to multimode situations for structured photonic environments. We note that issues with the single-mode

approximation have been recognised and discussed elsewhere[5,43], but that this approximation does not result in a breakdown of gauge invariance and does not therefore affect the results reported here. Within exact (non-truncated) models determining the dependence on the gauge parameter of light–matter entanglement, as well as averages of local light and matter observables such as photon number, is of experimental relevance and is important for applications. This too will be investigated in further work.

## Methods

**Lagrangians in different gauges**. The Coulomb-gauge Lagrangian is denoted $L_0$ and is given in Supplementary Note 1. More generally, the $\alpha$-gauge Lagrangian yielding the same correct equations of motion as $L_0$ is $L_\alpha = L_0 - d\chi_\alpha/dt$, where the function $\chi_\alpha$ is defined as

$$\chi_\alpha(t) = \alpha \int d^3x \mathbf{A} \cdot \mathbf{P}_{\text{mult}}, \tag{7}$$

$$P_{\text{mult},i}(\mathbf{x}) = -e \int_0^1 d\lambda\, r_j \delta_{ij}^{\text{T}}(\mathbf{x} - \lambda\mathbf{r}). \tag{8}$$

Here, $\mathbf{P}_{\text{mult}}$ denotes the usual multipolar transverse polarisation field. Latin indices denote spatial components and repeated indices are summed. This $\chi_\alpha$ is the generator of the unitary Power–Zienau–Woolley transformation, multiplied by $\alpha$. The $\alpha$-dependence of the Lagrangian can be understood as the underlying cause of the $\alpha$-dependence of the canonical momenta $\mathbf{p}_\alpha = \partial L_\alpha/\partial\dot{\mathbf{r}}$ and $\mathbf{\Pi}_\alpha = \delta L_\alpha/\delta\dot{\mathbf{A}}$.

**Derivation of cavity QED two-level model Hamiltonian**. Substituting Eqs. (1) and (2) into Eq. (3) yields the Hamiltonian written in terms of canonical operators $\mathbf{y}_\alpha$ as $H = H_{\text{m}}^\alpha + H_{\text{c}}^\alpha + V^\alpha$ where

$$H_{\text{m}}^\alpha = \frac{\mathbf{p}_\alpha^2}{2m} + V(\mathbf{r}), \tag{9}$$

$$H_{\text{c}}^\alpha = \frac{v}{2}\left(\mathbf{\Pi}_\alpha^2 + \omega^2 \mathbf{A}^2\right), \tag{10}$$

$$V^\alpha = \frac{e}{m}(1-\alpha)\mathbf{p}_\alpha \cdot \mathbf{A} + \alpha\widehat{\mathbf{d}} \cdot \mathbf{\Pi}_\alpha \\ + \frac{e^2}{2m}(1-\alpha)^2 \mathbf{A}^2 + \frac{\alpha^2}{2v}(\boldsymbol{\varepsilon} \cdot \widehat{\mathbf{d}})^2. \tag{11}$$

The Hamiltonian has a hybrid form between the Coulomb and multipolar gauges. Coulomb-gauge coupling terms are weighted by $1 - \alpha$ while multipolar-gauge coupling terms are weighted by $\alpha$. The interaction includes the quadratic "$\mathbf{A}^2$" and "$\widehat{\mathbf{d}}^2$" self-energy terms in addition to the linear coupling terms "$\mathbf{p}_\alpha \cdot \mathbf{A}$" and "$\widehat{\mathbf{d}} \cdot \mathbf{\Pi}_\alpha$". This approach is easily adapted to describe multimode fields and more than one dipole[36].

The first two eigenstates of the material bare energy $H_{\text{m}}^\alpha$ are denoted $|\epsilon_0^\alpha\rangle$ and $|\epsilon_1^\alpha\rangle$, and the projection onto this subspace is $P^\alpha = |\epsilon_0^\alpha\rangle\langle\epsilon_0^\alpha| + |\epsilon_1^\alpha\rangle\langle\epsilon_1^\alpha|$. The operator $H_{\text{m}}^\alpha$ admits the two-level truncation $H_{\text{m},2}^\alpha = P^\alpha H_{\text{m}}^\alpha P^\alpha = \epsilon_0 + \omega_{\text{m}}^\alpha \sigma_\alpha^+ \sigma_\alpha^-$, where $\omega_{\text{m}} = \epsilon_1 - \epsilon_0$, $\sigma_\alpha^+ = |\epsilon_1^\alpha\rangle\langle\epsilon_0^\alpha|$ and $\sigma_\alpha^- = |\epsilon_0^\alpha\rangle\langle\epsilon_1^\alpha|$. The eigenvalues $\epsilon_0$ and $\epsilon_1 = \omega_{\text{m}} + \epsilon_0$ corresponding to $|\epsilon_0^\alpha\rangle$ and $|\epsilon_1^\alpha\rangle$, respectively, are $\alpha$-independent because $H_{\text{m}}^\alpha = R_{\alpha\alpha'}H_{\text{m}}^{\alpha'}R_{\alpha\alpha'}^{-1}$. In practice, two-level model Hamiltonians are found by first defining the interaction Hamiltonian as $V_2^\alpha = V^\alpha(P^\alpha \mathbf{y}_\alpha P^\alpha)$ and then combining this interaction with the bare energies to obtain the total Hamiltonian

$$H_2^\alpha = P^\alpha H_{\text{m}}^\alpha P^\alpha + H_{\text{c}}^\alpha + V^\alpha(P^\alpha \mathbf{y}_\alpha P^\alpha). \tag{12}$$

If the interaction Hamiltonian $V^\alpha$ is linear in $\mathbf{r}$ and $\mathbf{p}_\alpha$ then the two-level model Hamiltonian can also be written $H_2^\alpha = P^\alpha H P^\alpha$. This is not the case for $H$ in Eq. (11) due to the "$\widehat{\mathbf{d}}^2$" term, which demonstrates the availability of different methods for deriving truncated models. Here, we adopt the approach most frequently encountered in the literature, and outline other methods in Supplementary Note 2.

We can now define an arbitrary-gauge two-level model associated with the Hamiltonian $H$ in Eq. (11) by using the definition (12). The projection $P^\alpha$ does not alter the "$\mathbf{A}^2$" and $H_{\text{c}}^\alpha$ terms of Eq. (11), because these terms depend on the cavity canonical operators only. Combining then gives the renormalised cavity energy $H_{\text{c}}^\alpha + e^2/2m(1-\alpha)^2\mathbf{A}^2 = \omega_\alpha(c_\alpha^\dagger c_\alpha + 1/2)$ with renormalised cavity frequency $\omega_\alpha = \omega\sqrt{1 + e^2(1-\alpha)^2/mv\omega^2}$. The $c_\alpha, c_\alpha^\dagger$ are cavity ladder operators of the renormalised energy satisfying $[c_\alpha, c_\alpha^\dagger] = 1$. In terms of these operators, the Hamiltonian $H_2^\alpha$ defined by Eq. (12) is given by Eq. (5).

**Method for comparing two-level model predictions**. A comparison of the predictions that different two-level models yield for an arbitrary observable requires

that we determine how a given physical state is represented within each two-level model. To this end, consider an observable $A$ with the property that both the exact representation $A$ and the two-level model representation $A_2^\alpha$ possess non-degenerate discrete spectra. The eigenvalues $a_n$ of $A$ and $a_{2,n}^\alpha$ of $A_2^\alpha$ are in one-to-one correspondence such that the eigenstates $|A_n\rangle$ and $\left|A_{2,n}^\alpha\right\rangle$ can be assumed to represent the same physical state. An arbitrary physical state can then be constructed via linear combination; the physical state $|\psi\rangle = \sum_n \psi_n |A_n\rangle, \sum_n |\psi_n|^2 = 1$ within the exact theory, is represented within the $\alpha$-gauge two-level model by $\left|\psi_2^\alpha\right\rangle = \sum_n \psi_n \left|A_{2,n}^\alpha\right\rangle$. A natural choice of observable $A$ for the purpose of representing states is the energy $A = H$, which we consider in Results.

The most accurate two-level model for the purpose of predicting the average $\langle\psi|O|\psi\rangle$ of an arbitrary observable $O$, which may or may not equal $A$, is found by selecting the gauge $\alpha$ for which the difference between the exact and two-level model prediction, $z^\alpha(O, \psi) = \left|\langle\psi|O|\psi\rangle - \left\langle\psi_2^\alpha\right|O_2^\alpha\left|\psi_2^\alpha\right\rangle\right|$, is minimised. Since two-level models are indispensable practical tools within cavity and circuit QED, it is important to ascertain which two-level models yield the best approximations of physical averages that are of interest in applications. In Results, the energy is considered, both to represent states ($A = H$) and as the observable of interest ($O = H$). The averages $\langle A_n|O|A_n\rangle$ are then nothing but the eigenvalues $E_n$ of $H$.

As an example illustrating how the relative accuracies of two-level models can be determined, let us consider the quantities $z^\alpha(O, G)$ where $G$ denotes the ground state of a composite cavity-charge system. The charge is assumed to be confined in all directions except the direction $\boldsymbol{\varepsilon}$ of the cavity mode polarisation. In this direction, it oscillates harmonically with bare frequency $\omega_{\text{m}}$. In the gauge specified by choosing $\alpha = \omega_{\text{m}}/(\omega_{\text{m}} + \omega)$, the matter oscillator can be described by ladder operators for which the interaction Hamiltonian takes number-conserving form[37]. The exact ground state $G$ is then the vacuum state of these modes, and the projection $P^{\text{JC}}$ onto the first two material levels in this gauge defines a two-level JCM with ground state $\left|G_2^{\text{JC}}\right\rangle = P^{\text{JC}}|G\rangle = |G\rangle$. It follows that $z^\alpha(O, G) = 0$ for all $O$ with $O_2^\alpha = P^\alpha O P^\alpha$. Thus, if the material system is a harmonic oscillator, it is possible to derive a JCM that is necessarily more accurate than any derivable QRM for finding ground-state averages.

**Fluxonium-$LC$ two-level model Hamiltonian**. The derivation in Supplementary Note 3 yields the $\alpha$-gauge fluxonium-$LC$ Hamiltonian

$$H = \frac{E_c}{e^2}[\xi_\alpha + (1-\alpha)\zeta]^2 + 2e^2 E_{\text{l}}\phi^2 \\ - E_{\text{J}}\cos(2e[\phi - \phi_{\text{ext}}]) + \frac{\zeta^2}{2C} + \frac{1}{2L}[\theta_\alpha + \alpha\phi]^2. \tag{13}$$

The fluxonium bare energy is defined as

$$H_{\text{m}}^\alpha = \frac{E_c}{e^2}\xi_\alpha^2 + 2e^2 E_{\text{l}}\phi^2 - E_{\text{J}}\cos(2e[\phi - \phi_{\text{ext}}]). \tag{14}$$

The projection onto the first two eigenstates of this operator is used along with $H$ in Eq. (13) to define a two-level model Hamiltonian in precisely the same way as in the cavity QED case. The final result is given in Eq. (6).

## Data availability

The data that support the findings of this study are available from the corresponding authors upon reasonable request.

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

## Acknowledgements

This work was supported by the UK Engineering and Physical Sciences Research Council, grant no. EP/N008154/1. We thank Zach Blunden-Codd for useful discussions.

## Author contributions

All authors contributed to all aspects of this work.

## Additional information

**Competing interests:** The authors declare no competing interests.

