## [Peer Review File · Nature Communications]

Reviewers' comments:

Reviewer #1 (Remarks to the Author):

In the manuscript entitled "Gauge ambiguities in QED: Jaynes-Cummings physics remains valid in the ultrastrong-coupling regime" by Stokes and Nazir, the authors discuss the correct treatment of a cavity-QED system and argue that the JC model can remain valid for strong coupling if the correct gauge is used in the derivation. The question of how to correctly describe these systems has been a topic of debate and controversy for a few decades. There is still disagreement on the correct theoretical treatment. This manuscript approaches the question from a new perspective. I therefore think that the manuscript is of interest for publication in Nature Communications.

However, there are a few points that I think should be addressed:

1) The authors generally plot the ground state and excited state energies. I think that it would be useful to plot the spectra that can be expected to be measured experimentally. In particular, Yoshihara et al. [Phys. Rev. A 95, 053824 (2017)] showed that the spectra evolve and undergo a few qualitative changes as the coupling strength is increased. All the spectra measured there were consistent with the predictions of the QRM. Can the authors comment on whether or how their results agree with those of Yoshihara et al.?

2) An important question in this area is whether a superradiance phase transition can occur in the Dicke model with strong coupling or the so-called no-go theorem will preclude this phase transition. Can the authors comment on what their approach predicts regarding this question?

3) In the introduction the authors mention that in the (conventional) JC model there is no entanglement in the ground state while in the QRM model there is entanglement for strong coupling. It was not clear to me whether the authors are claiming that in their approach (using a JC model) will lead to the prediction that there is no entanglement even for strong coupling. This point might need to be clarified.

4) While the manuscript is well written, there are a number of typos and small errors that need to be fixed. I suggest that the authors go over the manuscript carefully and try to fix these.

Reviewer #2 (Remarks to the Author):

In this manuscript the authors address the problem of the validity of some approximated models of matter-radiation interaction in the ultra-strong coupling regime, when a truncation of the atomic states is made (two-level approximation, in particular), for a general choice of the gauge. They also consider in detail a specific Circuit QED setup. As it is known, for example in the case of the Power-Zienau-Wolley (PZW) gauge transformation, the truncated (approximated) Hamiltonians are not unitarily equivalent and can then yield different predictions. In this paper, the authors find that, considering specific gauge choices, the JCM in a specific gauge can give, in the ultra-strong coupling regime, more accurate predictions than the QRM (for the physical quantities they evaluate). For some specific values of the parameter α specifying the gauge, they find that the Hamiltonian assumes a JCM form. In the Circuit QED case, they directly show the validity of a JC-gauge model in the ultra-strong regime.

In my opinion, this paper is physically sound and relevant for its field, and it deserves publication. It is also clearly written. There are however several points that, in my opinion, should be clarified and discussed before publication, also to show a general interest and relevance of the results obtained (relevant aspects for publication in Nature Communications).

- The authors consider only one field mode. Is this coherent with all considerations about gauge invariance, that usually refer to the complete field operators? Even if this point is briefly mentioned in the SM, it should be discussed in more detail, also in the main text of the paper. Also, do the authors expect to reach different conclusions in a multi-mode case?

- The fact that, when a two-level approximation is made, the results obtained using different gauges can be qualitatively different, is already well known in the literature, for example for the minimal and multipolar coupling schemes, even in weak coupling (see, for example, Ref. [43] of the manuscript). This should be clearly stated and discussed in the paper.

- The authors use a linear matter-radiation coupling. In nonrelativistic QED, the A^2 term is present (Coulomb gauge). The presence of this term, for example, is relevant for the equivalence of the minimal and the multipolar coupling schemes through the PZW gauge transformation. Also, one could reasonably expect that it is relevant in the strong and ultra-strong coupling regimes. Why the authors have not included this term in their Hamiltonian (see eq. (4))?

- Electrostatic interactions seem not involved in the general gauge transformation considered (they appear only in the Hamiltonian of matter). In the PWZ gauge transformation, dipolar electrostatic interactions (dipole-dipole or dipole-surface) are indeed involved; for example, in the multipolar scheme these interactions are included in the interaction of matter with the transformed transverse field (transverse displacement field), contrarily to the minimal coupling scheme, where they explicitly appear (separately from the transverse fields). This is very important for atom-atom and atom-surface dispersion interactions. Can the approach of the authors be applied to these cases also, and in more general contexts of cavity QED?

- Some points in Sec. III are not very clear, for example where Eq. (4) comes from, as well as the subsequent discussion, in particular what the quantities $\langle z^\alpha \rangle(O,g)$ (mentioned around the end of the section) are. I would also suggest the authors to mention in Sec. II the mathematical definition of the parameter α that fixes the gauge (it is mathematically defined only in SM section VIA).

- In Sec. IV, the comparison of energy shifts, fidelities and average photon numbers (ground and first excited state) between different two-level models and exact results is discussed. The conclusions of the authors on the accuracy of the various models considered are limited to these quantities, or are valid also for other physical quantities?

- The discussion on the plots in the SM section VID is not easy to follow and catch the main points; I would suggest the authors to organize it in a clearer form, stressing the main points.

In conclusion, my recommendation is that this paper can be published if the authors can successfully address all points raised above.

Reviewer #3 (Remarks to the Author):

The manuscript by Stokes and Nazir discusses the interplay between the choice of gauge and the two-level approximation for a single atom or artificial atom coupled to a single mode cavity. The manuscript makes two main novel claims. Firstly, that there exists a gauge (referred to as the Jaynes-Cummings gauge) in which the matter-light coupling is automatically of the Jaynes-Cummings form,

i.e. there are no counter-rotating terms. The second claim is that this gauge is in some senses the "best" gauge in which to work, at least regarding ground state properties. The first of these claims is clearly demonstrated, and this on its own would I think already be sufficient grounds for publication in Nature Communications: this result is a new and important observation about the structure of the matter light coupling, and provides a dramatic example of the gauge dependence of entanglement measures. The second claim is less clear and, as discussed below, it would help if there were some further discussion of this point. However, overall I believe the manuscript should be published in Nature Communications after the authors address the points below.

The essential question noted above is whether the results here are enough to claim the JC gauge is the best gauge. In most of the manuscript this is stated reasonably clearly as a statement about the ground state, or perhaps the lowest two levels. In the abstract, this statement is made ever more strongly: this says the Jaynes-Cummings model is often "more accurate": I am not sure the evidence in the manuscript supports this statement without the qualifier "for the ground state".

As the authors are aware, there is another recent preprint, Ref. 35, which states a different answer, that the Dipole gauge is best (from a consideration of only two gauges). In that paper, for different models, they in fact show the dipole gauge captures the evolution of many energy levels of the Hamiltonian very well. In the current manuscript, this point is partially addressed by the comparison of the circuit QED models for different values of δ , where it becomes clear that for small δ , the dipole gauge is indeed better for capturing higher lying levels.

What would be helpful is if there were clearer statements that could be made about why the behavior is as it is: i.e. why the JC model is best for the ground state but the dipole gauge best for higher levels. Further, it would be very helpful to the community if the authors could consider applying their general gauge to one of the same models presented in Ref. 35, allowing a like for like comparison. However, it may be that this is beyond the scope of the current paper.

Expanding on the above, I also wonder whether there could be more discussion explaining why it is that the Jaynes-Cummings model works so well for the ground state. The statement on page 5 that this is because "the JC-gauge interpolates between the flux and charge gauges" seems rather vague -- the fact that one gauge overestimates and the other underestimates the energy means there will be a better result between these points, but not that the JC gauge will be near the optimum point, as it seems always to be. It seems from elsewhere in the manuscript, for the harmonic matter model, there is a clear reason, namely that because there are no counter-rotating terms in the matter-light coupling, the ground state of the full model and the ground state of the two-level approximated models must match. i.e. number conservation protects the ground state. In more general models, there presumably are matter-light coupling terms involving transitions to higher atomic states, and these do not necessarily simplify at the same JC point that the ground state coupling does. (Presumably there can actually be other special points, where higher transitions between the ground

state and n th excited state couple to light with a simple JC form). Despite this complication, there may be an argument to be made here about the size of such terms as being smaller in perturbation theory. This could provide a more convincing explanation why the JC gauge is generally best for the ground state.

Regarding the JC vs dipole gauge, Ref. 35 makes the argument that the dipole gauge is best because the form of coupling coefficients means transitions from the ground state to higher lying levels have a weaker coupling to light. I wonder if this statement can be studied in the general α gauge: can one ask how the ground-state to higher excited matter states scales with energy, and how this behaves in each gauge. Answering this might help clarify which gauge is best under which circumstances.

It is not necessary for all the above to be addressed for publication, but I suggest the authors at least consider whether these points can be addressed as this would increase the significance of their work.

In addition to the above, there are a number of presentational points the authors should consider.

* There are quite a few places where I am unclear about the logic used in dividing material between the main text the supplemental material. While I agree that derivations can be safely relegated to the supplemental material there are some points I might have expected to see in the main text. These include: (1) an explicit form of χ_α for the cavity QED case (i.e. Eq. 21 along with the definition of P_{mult}); (2) The Hamiltonian for the circuit QED case, perhaps Eq. 49 for the general case. I would suggest the authors consider these specific suggestions, and also review to ensure the supplemental material is really supplemental, i.e. whether one can understand their results from the main text alone.

* Regarding the list of references on gauge dependence of two-photon transitions (currently 28-32) there are other papers on this topic that it might be appropriate to cite:

W. E. Lamb, Jr., Phys. Rev. 85, 259 (1952).

K.-H. Yang, Ann. Phys. (N.Y.) 101, 62 (1976).

J. J. Forney, A. Quattropani, and F. Bassani, Nuovo Cimento 37B, 78 (1977).

D. H. Kobe, Phys. Rev. Lett. 40, 538 (1978)

* In figure 2 onwards, I found all these figures had font sizes slightly too small to comfortably read, and that the lines perhaps had too narrow a linewidth.

* In Figure 3, I was unclear why the c-JCM was shown, but the f-JCM was not shown.

* I found the discussion of how to compare accuracy of gauges on page 4 unclear. The discussion suggests one needs to choose two things: (1) an observable A , that is used to produce an eigenbasis, and (2) an observable O whose value is calculated. It is not clear whether both these choices have an effect. From the rest of the manuscript, my understanding is that once the truncation of the matter to a two-level system has been made, this is a truncation of the Hilbert space. As such, it would seem that while it clearly matters what observable O is compared, the choice of A is irrelevant, since the eigenstate resolution is a completeness relation in the restricted Hilbert space.

Further, the wording in this paragraph seems to say that in the following the choice $A=H$ is made, and does not discuss what observable O is being considered. The results however suggest that it is $O=H$ that is chosen for almost all cases.

I may have misunderstood this paragraph, however if this is the case, I expect many readers would also misunderstand this.

* At the end of section III (on page 4), there are two slightly confusing points of notation. Firstly $z^\alpha(O,g)$ is used to refer to the ground state, however elsewhere the ground state is being labeled by an upper case G . Secondly, the value of α for the Jaynes-Cummings gauge is labeled as α_g . In all other places, this is referred to as α_{JC} .

* On page 5, there is a statement about predicting dressed energies " $N>E$ ". This notation is unclear, since neither N nor E are defined. It is repeated several times and is unclear each time. I believe the authors mean this to refer to higher excited states, but since it's not clear what N is, this isn't obvious.

* In the supplemental material section A, there is a switch of notation from $g(x',x)$ in Eq. 11 to g_T in the text following it. Since $g(x,x')$ is already defined as satisfying the gauge fixing condition, it seems g and g_T refer to the same thing. If so, the notation should be consistent. If they are not the same things, this needs better explanation.

* I am confused by the discussion of the Gauge fixing condition $C_2=0$. It appears that for $C_2=0$ to be satisfied, it is necessary that Eq. 7 is satisfied, as this seems to be required in order that Eq. 11 satisfies Eq. 10. However, it would seem Eq. 17 does not in general satisfy Eq. 7. It only satisfies it when $\alpha=1$, for other values it does not seem to be normalized as required to satisfy Eq. 7. Since $C_2=0$ is a gauge fixing condition, this doesn't seem to actually be a problem for the main results, however the presentation seems to be written as if all choices of g considered lead to C_2 being zero.

* The discussion in supplemental section C is not clear. The first half of this section seems all to be predicated on the consequences of assuming that the TRK sum rule is satisfied for the two levels of the matter system. I do not understand why such an assumption would be considered. In most work I am aware of, the TRK sum rule is used on the total sum over all levels, and is then used as to provide an inequality for the transitions between the lowest two levels. i.e., for transitions from the ground state, one can know all terms are positive on the left hand side of Eq. 51, so each term individually must be less than Eq. 51. I am not aware of other works starting from the premise that in the two-level approximation the remaining term in the sum must be equal to the TRK sum.

* There is a typo on page 18, "differewnt"

We thank the editors for consideration of our manuscript and we thank the reviewers for useful and constructive comments. In addressing these comments we believe the manuscript has improved significantly. There are no comments that we strongly contend and we have endeavoured to respond to each with an appropriate level of detail.

Our responses are colour-coded, both in the manuscript and below. (Green) Reviewer 1, (Blue) Reviewer 2, (Red) Reviewer 3.

In some places we have included a brief annotation in the manuscript in *italics* to explain a notational change. In some places we have struck-through parts of text rather than omit it. This is so reviewers can compare what we wish to be deleted with the replacements we have made. The manuscript now includes more detail and additional analysis, and has therefore lengthened. Several new references have also been added.

Reviewer comments

Reviewer 1 (Remarks to the Author):

In the manuscript entitled "Gauge ambiguities in QED: Jaynes-Cummings physics remains valid in the ultrastrong-coupling regime" by Stokes and Nazir, the authors discuss the correct treatment of a cavity-QED system and argue that the JC model can remain valid for strong coupling if the correct gauge is used in the derivation. The question of how to correctly describe these systems has been a topic of debate and controversy for a few decades. There is still disagreement on the correct theoretical treatment. This manuscript approaches the question from a new perspective. I therefore think that the manuscript is of interest for publication in Nature Communications.

However, there are a few points that I think should be addressed:

1) The authors generally plot the ground state and excited state energies. I think that it would be useful to plot the spectra that can be expected to be measured experimentally...

Although spectroscopy probes transition frequencies (eigenvalue differences) we opted to plot the eigenvalues themselves. The reason is that it is possible for approximate models to fortuitously accurately predict the difference between two energies by incurring equal error in both energies, despite being inaccurate for the energies themselves. In contrast the energies themselves give an unmistakable representation of the accuracy of the approximate models.

On the other hand we agree that it would be useful to plot directly measurable spectra. For a fluxonium-LC system spectroscopic experiments are able to give data for transition energies as the external flux is varied (e.g. Ref. [2]). Originally we plotted the first two eigenvalues in the large detuning regime (also experimentally relevant) as a function of external flux (Fig. 4). Higher levels tend to be inaccurately

predicted by two-level models in this regime (as shown and explained in SM).

To give a more direct representation of the measurable spectra we have now changed Fig. 4 to show the first transition instead of the first level's energy, which we have instead included as an inset. The actual information presented stays the same, but the change now gives a direct representation of the experimentally accessible energy.

1) (continued...) In particular, Yoshihara et al. [Phys. Rev. A 95, 053824 (2017)] showed that the spectra evolve and undergo a few qualitative changes as the coupling strength is increased. All the spectra measured there were consistent with the predictions of the QRM. Can the authors comment on whether or how their results agree with those of Yoshihara et al.?

We focus on the first two-levels of a fluxonium-LC system and envisage a situation in which any experimental data would be fitted to the non-truncated theory to extract the model parameters, as is done in Ref. 2 for example. These extracted parameters can in turn be used in any associated two-level model. In contrast the authors of Phys. Rev. A 95, 053824 (2017) (Y) consider for their circuit QED system a fit of their data directly to a truncated model, specifically a QRM. This is the way they extract parameters of their truncated model. That this is possible is in no way inconsistent with our findings. Thus, there is certainly no contradiction between our results and Y.

More directly relevant experimental work than Y is given for our fluxonium-LC system in Ref. 2. We now explain these details in the manuscript at the end of Sec. IV in the new subsection entitled "Discussion". We state that our results are consistent with experiments performed to date, and in particular they do not disagree with Y. We explain that the JC gauge two-level model would give better agreement with experiment in the regimes that we consider for the lowest two states if data is fitted to the non-truncated model (as for example is done in Ref. 2).

To see whether our results might be relatable to those of Y we mapped our flux-gauge QRM on to the form used there. We have included this calculation in an accompanying PDF. We found that the parameters of the QRM in Y are non-trivial functions of the fluxonium-LC system parameters. In Y coupling and frequency parameters are fixed while the bias is varied. In contrast, if we vary external flux then the bias of our flux-gauge QRM varies, but so too do the other parameters in the Hamiltonian. We can of course directly vary the bias in the flux gauge QRM (rather than varying external flux) and hold the other parameters fixed, but doing this just amounts to a trivial reproduction of the results of Y. Varying our flux-gauge QRM's bias while holding the other parameters fixed does not describe a fluxonium-LC experiment in which external flux is varied. We are unable to write the non-truncated theory for the fluxonium-LC system in terms of a bias which can be varied while the other parameters are fixed. In this sense the fluxonium-LC system we consider is a different physical system to that in Y.

In Y the QRM they consider must be a truncation of an underlying non-truncated theory (though this description is not given in Y). It must also be possible to fit their data to the relevant non-truncated theory, as a way to extract the required parameters. If these parameters were then used in their QRM either the QRM would

be found to deviate from the experiment, or, their QRM and the data and the non-truncated theory would all match, which would show that for their system the truncation used is very accurate.

In any eventuality the results of Y do not really imply anything about ours and vice versa. What we are able to do (and have done) is compare at maximal frustration our results for the flux-gauge QRM to those of Y at zero bias using the parameters they specify. We found that the two sets of results do agree as they must. This amounts to a consistency check that ourselves and the authors of Y have produced the same numerical diagonalisation of the QRM at the single point in parameter space where our flux-gauge QRM coincides with the QRM in Y.

2) An important question in this area is whether a superradiance phase transition can occur in the Dicke model with strong coupling or the so-called no-go theorem will preclude this phase transition. Can the authors comment on what their approach predicts regarding this question?

We recognise that this topic is both interesting and important, and we have given it thought, but the topic seems to be quite involved. A precursor to the occurrence of a phase transition in the Dicke model is the violation of the "no-go theorem" for exponential closure of the ground state energy gap in the Rabi model (see for example Ref. 41). This theorem puts a bound on the Coulomb-gauge Rabi model's coupling strength using the TRK sum rule. We note that in contrast to the Coulomb and dipole gauges the number-conserving and counter-rotating terms have different coupling strengths in the α -gauge. We can (and have) used the TRK sum-rule to bound them. The bound generally depends on the material potential as in Ref. 41 for the dipole gauge. This can actually be concluded immediately because the α -gauge always has a component of dipole gauge coupling (weighted by α). This means that strictly speaking the no-go theorem holds in general, i.e., independent of the material potential, only in the Coulomb gauge. We have explained this in the manuscript after Eq. (7).

An important point however is that it is not really clear to what extent the different behaviours regarding the phase transition, i.e., the question; is there one or isn't there, actually depends on the two-level truncation. Based on our preliminary investigations the occurrence of phase transition seems to depend largely on whether bare or renormalised model parameters are used in the analysis.

We think that this warrants its own dedicated investigation using the arbitrary gauge approach. We think that an attempt to discuss this topic in detail here would not be adequate and would most likely achieve little more than distracting from the main points we want to make regarding two-level models. Our analysis and conclusions also hold independently of this topic. For this reason, and to not overload the manuscript we have included minimal details of it, but we make it clear that we want to investigate it in the future. Within the conclusions section we have now stated that a thorough investigation is left for further work. We have added more detailed comments regarding the topic after Eq. (7) and have added relevant references.

3) In the introduction the authors mention that in the (conventional) JC model there is no entanglement in the ground state while in the QRM model there is entanglement for strong coupling. It was not clear to me whether the authors are claiming that in their approach (using a JC model) will lead to the prediction that there is no entanglement even for strong coupling. This point might need to be clarified.

In bullet point two towards the end of the introduction this point has now been clarified.

4) While the manuscript is well written, there are a number of typos and small errors that need to be fixed. I suggest that the authors go over the manuscript carefully and try to fix these.

We hope now that all typos and problems with notation have been fixed.

Reviewer 2 (Remarks to the Author):

In this manuscript the authors address the problem of the validity of some approximated models of matter-radiation interaction in the ultra-strong coupling regime, when a truncation of the atomic states is made (two-level approximation, in particular), for a general choice of the gauge. They also consider in detail a specific Circuit QED setup. As it is known, for example in the case of the Power-Zienau-Wolley (PZW) gauge transformation, the truncated (approximated) Hamiltonians are not unitarily equivalent and can then yield different predictions. In this paper, the authors find that, considering specific gauge choices, the JCM in a specific gauge can give, in the ultra-strong coupling regime, more accurate predictions than the QRM (for the physical quantities they evaluate). For some specific values of the parameter α specifying the gauge, they find that the Hamiltonian assumes a JCM form. In the Circuit QED case, they directly show the validity of a JC-gauge model in the ultra-strong regime.

In my opinion, this paper is physically sound and relevant for its field, and it deserves publication. It is also clearly written. There are however several points that, in my opinion, should be clarified and discussed before publication, also to show a general interest and relevance of the results obtained (relevant aspects for publication in Nature Communications).

- The authors consider only one field mode. Is this coherent with all considerations about gauge invariance, that usually refer to the complete field operators? Even if this point is briefly mentioned in the SM, it should be discussed in more detail, also in the main text of the paper. Also, do the authors expect to reach different conclusions in a multi-mode case?

This is a reasonable point; the validity of the single-mode approximation is relevant to ultrastrong coupling and has recently been investigated in the context of cavity QED [Ref. 43]. However, the single-mode approximation does not affect the gauge-invariance of the theory (single mode gauge-transformations remain unitary) and

there is no reason to believe that the conclusions we draw regarding two-level models would be different in a multi-mode description. In this sense the single-mode treatment is coherent with all considerations about gauge invariance. Also, single-mode treatments of a fluxonium-LC system have been found to be in good agreement with experiment [Ref. 2]. For simplicity and to best compare with the extensive literature on the QRM and JCM we consider the single-mode case. We have added a comment regarding the single-mode treatment before Eq. (2) which should clarify this point.

- The fact that, when a two-level approximation is made, the results obtained using different gauges can be qualitatively different, is already well known in the literature, for example for the minimal and multipolar coupling schemes, even in weak coupling (see, for example, Ref. [43] of the manuscript). This should be clearly stated and discussed in the paper.

We have added a comment regarding this point in the second to last paragraph of section 3. We have also expanded and reordered SM C, which discusses these points in detail.

- The authors use a linear matter-radiation coupling. In nonrelativistic QED, the A^2 term is present (Coulomb gauge). The presence of this term, for example, is relevant for the equivalence of the minimal and the multipolar coupling schemes through the PZW gauge transformation. Also, one could reasonably expect that it is relevant in the strong and ultra-strong coupling regimes. Why the authors have not included this term in their Hamiltonian (see eq. (4))?

The A^2 term is actually present in all of the Hamiltonians and is weighted by $(1-\alpha)^2$. In the Coulomb gauge ($\alpha=0$) the usual A^2 term emerges. We have now made this point much clearer by explicitly giving the full Hamiltonian in terms of canonical operators in Eq. (5). We have reordered the section entitled "non-equivalent two-level models" to explain better where Eq. (4) (now Eq. (7)) comes from. This should in addition explain where the self-energy terms including the A^2 term arise in our treatment.

- Electrostatic interactions seem not involved in the general gauge transformation considered (they appear only in the Hamiltonian of matter). In the PWZ gauge transformation, dipolar electrostatic interactions (dipole-dipole or dipole-surface) are indeed involved; for example, in the multipolar scheme these interactions are included in the interaction of matter with the transformed transverse field (transverse displacement field), contrarily to the minimal coupling scheme, where they explicitly appear (separately from the transverse fields). This is very important for atom-atom and atom-surface dispersion interactions. Can the approach of the authors be applied to these cases also, and in more general contexts of cavity QED?

This is an interesting point. Since we consider only a single dipole direct static interactions between different dipoles do not feature. The approach can be applied in the case of multi-dipole systems (and multi-mode systems). For example, we

recently used an arbitrary gauge approach to model a conventional atomic physics system consisting of two dipoles weakly coupled to the free radiation field [Ref. 39]. There is also no reason that the approach cannot be applied in more general cavity QED settings and atom-surface interactions, and this would be interesting to look at as part of ongoing work. We have now added a brief statement in section II after Eq. (5) about the applicability of our approach. In the conclusions section where we note that the extension to multi-mode systems is yet to be performed we have added the clarifier "for structured photonic environments".

- Some points in Sec. III are not very clear, for example where Eq. (4) comes from, as well as the subsequent discussion, in particular what the quantities $z^\alpha(O,g)$ (mentioned around the end of the section) are. I would also suggest the authors to mention in Sec. II the mathematical definition of the parameter α that fixes the gauge (it is mathematically defined only in SM section VIA).

Section III has been reorganised and more details of the derivation of the Hamiltonian are given. It should now be clear where Eq. (4) (now Eq. (7)) comes from having added text around what is now Eq. (5) and subsequently around what is now Eq. (7). We have also added Eq. (6) to show clearly how the two-level model is defined. Also, the parameter α is now defined in Eq. (1) (this change is in red in the manuscript, as the inclusion of this equation was explicitly suggested by reviewer 3). The notation regarding the quantities $z^\alpha(O,g)$ has been changed (as also suggested by reviewer 3) so it should now be clear that these represent differences between non-truncated and truncated theoretical predictions in the ground state.

- In Sec. IV, the comparison of energy shifts, fidelities and average photon numbers (ground and first excited state) between different two-level models and exact results is discussed. The conclusions of the authors on the accuracy of the various models considered are limited to these quantities, or are valid also for other physical quantities?

One cannot claim with certainty, which two-level model will be the most accurate for predictions of an arbitrary observable in an arbitrary state. However, from the relative accuracies of the various model predictions for the lowest energy states (fidelities and energy values) it is reasonable to suppose that the relative accuracies will be much the same for predictions of other observables within these low energy states. For example, since according to the fidelities calculated the JC-gauge seems to give the best representation of the ground state one might naturally expect it gives the most accurate predictions for ground state averages of an arbitrarily selected observable. This cannot be known for sure however without an analysis of the effect of the two-level approximation on the observable in question, or simply a direct comparison between the ground state predictions of the various two-level models and the non-truncated theory for the chosen observable. We have added a remark explaining this point after Fig. 4. The example we give of an observable other than the energy is photon number (discussed in SM). In this case the JC gauge two-level model does yield the most accurate ground state averages.

- The discussion on the plots in the SM section VID is not easy to follow and catch the main points; I would suggest the authors to organize it in a clearer form, stressing the main points.

We have added a paragraph at the beginning of this section which summarises the main points. We have also reworded some of the remaining parts, and made further additions, to make them clearer. We have added a new section to the SM (section D) analysing the non-truncated system in terms of an effective Hamiltonian. This analysis is intended to complement the already present section (now section E) in which further results are given. We have accordingly retained the division of section E into three subsections based on the detuning parameter. We have summarised the results of each subsection at the start, and have stated what each graph in the subsection shows. We have made additions to the main text in response to comments of the other reviewers, which explain the relation of our results to Ref. 41, which link to the new SM section D and to SM E. With the new additions we hope it is now straightforward to extract the main conclusions and points.

In conclusion, my recommendation is that this paper can be published if the authors can successfully address all points raised above.

Reviewer 3 (Remarks to the Author):

The manuscript by Stokes and Nazir discusses the interplay between the choice of gauge and the two-level approximation for a single atom or artificial atom coupled to a single mode cavity. The manuscript makes two main novel claims. Firstly, that there exists a gauge (referred to as the Jaynes-Cummings gauge) in which the matter-light coupling is automatically of the Jaynes-Cummings form, i.e. there are no counter-rotating terms. The second claim is that this gauge is in some senses the "best" gauge in which to work, at least regarding ground state properties. The first of these claims is clearly demonstrated, and this on its own would I think already be sufficient grounds for publication in Nature Communications: this result is a new and important observation about the structure of the matter light coupling, and provides a dramatic example of the gauge dependence of entanglement measures. The second claim is less clear and, as discussed below, it would help if there were some further discussion of this point. However, overall I believe the manuscript should be published in Nature Communications after the authors address the points below.

The essential question noted above is whether the results here are enough to claim the JC gauge is the best gauge. In most of the manuscript this is stated reasonably clearly as a statement about the ground state, or perhaps the lowest two levels. In the abstract, this statement is made ever more strongly: this says the Jaynes-Cummings model is often "more accurate": I am not sure the evidence in the manuscript supports this statement without the qualifier "for the ground state".

This is a reasonable point; a qualifier referring to the lowest two energy levels has now been added to the abstract. This has made the abstract less vague. We believe that on the whole, there is enough evidence in the manuscript to warrant the statement we have made referring to the first two-levels. We concede that occasionally the flux gauge QRM does perform better than the JC-gauge for the first

excited state for large enough coupling and certain external fluxes provided δ is small. However, in many regimes, particularly large detuning regimes relevant to experiments, the JC gauge two-level model seems to be best for the first excited state as well as the ground state. In the main text where the various graphs are analysed at the very end of section IV we have stated that there is an exception where the flux-gauge QRM is better for the excited state. We have also clarified this point in the conclusions section. We believe that now throughout the manuscript all statements regarding the relative performance of the two-level models are appropriately precise, and are justified given the results.

As the authors are aware, there is another recent preprint, Ref. 35, which states a different answer, that the Dipole gauge is best (from a consideration of only two gauges). In that paper, for different models, they in fact show the dipole gauge captures the evolution of many energy levels of the Hamiltonian very well. In the current manuscript, this point is partially addressed by the comparison of the circuit QED models for different values of δ , where it becomes clear that for small δ , the dipole gauge is indeed better for capturing higher lying levels.

What would be helpful is if there were clearer statements that could be made about why the behavior is as it is: i.e. why the JC model is best for the ground state but the dipole gauge best for higher levels. Further, it would be very helpful to the community if the authors could consider applying their general gauge to one of the same models presented in Ref. 35, allowing a like for like comparison. However, it may be that this is beyond the scope of the current paper.

We think that a full comparison with models in Ref. 35 (now Ref. 41) would unfortunately be beyond the scope of the present paper, but we keep this comparison in mind for further work, where, we intend to look at questions regarding phase transitions etc using our formalism. Ref. 41 appeared as a preprint only a day before our own manuscript, so we did not know which systems they had considered until our manuscript was already complete and all results had already been obtained. However, we have since spoken to the authors and we recognise that the results of Ref. 41 are complementary to our own, and should be discussed more. We have rectified this oversight by including early on in the manuscript (in the introduction) a summary of the main point in Ref. 41 relevant to our work; this is that the dipole gauge (flux gauge) two-level model performs better for higher energy levels for small enough detuning δ .

We have made extensive additions which explain why two-level models of certain gauges perform better in certain regimes. We have added a whole new SM section (section D) which derives an effective Hamiltonian allowing an understanding of how two-level truncations perform in different gauges in different regimes. This includes techniques already used in Ref. 41 among other methods, notably those of Ref. 54. The new SM D thereby extends some of the analysis in Ref. 41.

We explain why the dipole (flux) gauge generally does better for higher levels with small detuning. We offer an explanation for the breakdown of the charge, flux and JC gauge two-level models in the regimes that they breakdown. We explain why the JC gauge gives a good representation of the ground state. We summarise the main

points of this analysis in the main text towards the end of Sec. IV by including a new Discussion subsection. The details of the analysis are given in SM D.

Expanding on the above, I also wonder whether there could be more discussion explaining why it is that the Jaynes-Cummings model works so well for the ground state. The statement on page 5 that this is because "the JC-gauge interpolates between the flux and charge gauges" seems rather vague --- the fact that one gauge overestimates and the other underestimates the energy means there will be a better result between these points, but not that the JC gauge will be near the optimum point, as it seems always to be. It seems from elsewhere in the manuscript, for the harmonic matter model, there is a clear reason, namely that because there are no counter-rotating terms in the matter-light coupling, the ground state of the full model and the ground state of the two-level approximated models must match. i.e. number conservation protects the ground state. In more general models, there presumably are matter-light coupling terms involving transitions to higher atomic states, and these do

not necessarily simplify at the same JC point that the ground state coupling does. (Presumably there can actually be other special points, where higher transitions between the ground state and n th excited state couple to light with a simple JC form). Despite this complication, there may be an argument to be made here about the size of such terms as being smaller in perturbation theory. This could provide a more convincing explanation why the JC gauge is generally best for the ground state.

We agree with this analysis. As noted by the referee the reason we included the discussion of the coupled oscillator case was to demonstrate that in that case it is clear why the JCM gives a better ground state than the QRM. As the referee notes for a general multi-level system virtual transitions to higher levels cannot be eliminated by the same choice of gauge as is necessary to eliminate counter-rotating terms between the first two-levels. However, as the referee also conjectured it is indeed possible to eliminate counter-rotating terms associated with any one atomic transition at a time, i.e., one needn't necessarily choose to eliminate the counter-rotating terms for the first transition. The reason all counter-rotating terms can be eliminated in the case of either a material oscillator or a material 2LS is that in these cases there is only one material frequency involved in the problem.

Virtual transitions out of the ground state can be taken to quantify the deviation of the ground state from the vacuum $|0,0\rangle$. In the JC gauge these deviations are smaller in the sense that only transitions to higher states are present - those to the first state have been eliminated. Thus, in the JC gauge the vacuum is closer to the ground state than in other gauges. Moreover, the JC gauge two-level model ground state is the vacuum. If transitions to higher levels out of the vacuum are small then the two-level approximation will give a good approximation of the ground state, precisely as it seems to in the JC gauge. Therefore, as we think the referee is arguing, we agree that a demonstration that counter-rotating transitions to higher levels are small would explain why the JC-gauge always comes close to the optimum two-level approximation for the ground state. This demonstration has essentially now been given in SM D and summarised in the main text (section IV Discussion). We also analyse in SM D the behaviour of the JC-gauge parameter α_{JC} as a function

of detuning and coupling to better understand how it mixes the charge and flux gauges. We thereby obtain an idea of how the JC-gauge two-level model changes in different regimes.

Regarding the JC vs dipole gauge, Ref. 35 makes the argument that the dipole gauge is best because the form of coupling coefficients means transitions from the ground state to higher lying levels have a weaker coupling to light. I wonder if this statement can be studied in the general α gauge: can one ask how the ground-state to higher excited matter states scales with energy, and how this behaves in each gauge. Answering this might help clarify which gauge is best under which circumstances.

To some extent this analysis is now performed in SM D (see above two comments). Summarising remarks concerning this have been added to the main text (Sec. IV Discussion) and SM D includes more details. The general α -gauge expressions we obtained were not especially illuminating, but we know that the general α gauge mixes flux and charge couplings weighted by α and $1-\alpha$ respectively. We have therefore focused on these gauges and on the JC gauge in SM D. In so doing we build up a reasonably comprehensive picture of the behaviour of the different two-level models in different regimes.

It is not necessary for all the above to be addressed for publication, but I suggest the authors at least consider whether these points can be addressed as this would increase the significance of their work.

In addition to the above, there are a number of presentational points the authors should consider.

* There are quite a few places where I am unclear about the logic used in dividing material between the main text the supplemental material. While I agree that derivations can be safely relegated to the supplemental material there are some points I might have expected to see in the main text. These include: (1) an explicit form of χ_α for the cavity QED case (i.e. Eq. 21 along with the definition of P_{mult}); (2) The Hamiltonian for the circuit QED case, perhaps Eq. 49 for the general case. I would suggest the authors consider these specific suggestions, and also review to ensure the supplemental material is really supplemental, i.e. whether one can understand their results from the main text alone.

Taking the referees suggestions on board we have included the suggested parts of the supplementary material in the main text, giving Eqs. (1) and (8) as new equations in the text. It should hopefully now be possible to read through the main text without ever having to refer to the supplementary material to obtain the main points and conclusions drawn within the manuscript. SM contains longer derivations and more detailed analyses.

* Regarding the list of references on gauge dependence of two-photon transitions (currently 28-32) there are other papers on this topic that it might be appropriate to

cite:

W. E. Lamb, Jr., Phys. Rev. 85, 259 (1952).

K.-H. Yang, Ann. Phys. (N.Y.) 101, 62 (1976).

J. J. Forney, A. Quattropani, and F. Bassani, Nuovo Cimento 37B, 78 (1977).

D. H. Kobe, Phys. Rev. Lett. 40, 538 (1978)

References have been added.

* In figure 2 onwards, I found all these figures had font sizes slightly too small to comfortably read, and that the lines perhaps had too narrow a linewidth.

Figures have been remade with larger text and thicker lines, so they should be easier to read.

* In Figure 3, I was unclear why the c-JCM was shown, but the f-JCM was not shown.

The reason is only that the flux-gauge QRM is already very inaccurate in this regime (as can be seen in the graphs) and the flux gauge JCM is even more inaccurate. We could include the flux gauge JCM, but rather than clutter the graphs we opted to state in the caption that the flux-gauge JCM is extremely inaccurate and is therefore not shown.

* I found the discussion of how to compare accuracy of gauges on page 4 unclear. The discussion suggests one needs to choose two things: (1) an observable A , that is used to produce an eigenbasis, and (2) an observable O whose value is calculated. It is not clear whether both these choices have an effect. From the rest of the manuscript, my understanding is that once the truncation of the matter to a two-level system has been made, this is a truncation of the Hilbert space. As such, it would seem that while it clearly matters what observable O is compared, the choice of A is irrelevant, since the eigenstate resolution is a completeness relation in the restricted Hilbert space.

We have reworded this part of the manuscript to make it clearer. To compare predictions one must be able to represent the same physical state by Hilbert space vectors in each model. Given an arbitrary observable and projection, the projection of an eigenvector of the observable is not generally an eigenvector of the projected observable. Establishing correspondence between states of the non-truncated and projected theories is therefore not completely trivial. The issue concerns establishing bijective correspondence between states of both models. For example, the dipole moment in the non-truncated model has continuous unbounded spectrum, but the dipole moment in the truncated theory is represented by σ_x , which has discrete binary spectrum. Moreover, in the full Hilbert space both operators are highly degenerate since they act in the material part only. If A were chosen as the dipole moment it would not be clear how to identify the representation of a given eigenstate of A within the truncated theory given its representation in the non-truncated theory. This is the purpose of identifying an A and A_2 with discrete non-

degenerate spectrum. We can then be sure that the eigenvalues a_n of A and its two-level counterpart are in one-to-one correspondence, and this provides a unique (natural) correspondence between eigenstates of the truncated and non-truncated theories. We do indeed use $A=H$ for this purpose, which is a natural choice. These points have been clarified at the relevant part of the manuscript.

Further, the wording in this paragraph seems to say that in the following the choice $A=H$ is made, and does not discuss what observable O is being considered. The results however suggest that it is $O=H$ that is chosen for almost all cases.

We have also clarified this together with the above mentioned changes. It should now be clear that we use $A=H$ and $O=H$.

I may have misunderstood this paragraph, however if this is the case, I expect many readers would also misunderstand this.

* At the end of section III (on page 4), there are two slightly confusing points of notation. Firstly $z^\alpha(O,g)$ is used to refer to the ground state, however elsewhere the ground state is being labeled by an upper case G . Secondly, the value of α for the Jaynes-Cummings gauge is labeled as α_g . In all other places, this is referred to as α_{JC} .

We have changed the notation so that these issues should now have been fixed. We have replaced g with G in z^α and elsewhere, and the use of notation α_g is not necessary so it has now been eliminated.

* On page 5, there is a statement about predicting dressed energies " $N>E$ ". This notation is unclear, since neither N nor E are defined. It is repeated several times and is unclear each time. I believe the authors mean this to refer to higher excited states, but since it's not clear what N is, this isn't obvious.

Notation has been changed so that it should now be clear that G and E are the first two energies of the overall system and E_n is used for higher energies $E_n>E$. ϵ_n refers to energies of the material system.

* In the supplemental material section A, there is a switch of notation from $g(x',x)$ in Eq. 11 to g_T in the text following it. Since $g(x,x')$ is already defined as satisfying the gauge fixing condition, it seems g and g_T refer to the same thing. If so, the notation should be consistent. If they are not the same things, this needs better explanation.

g and g_T are actually not the same thing, and this is an important point. Extensive clarification of this has been added (see response to comment below). In Eq. (11) (now Eq. (19)) the transverse vector potential under the integral picks out only the transverse part of g , i.e., the integral of the dot product of a transverse and longitudinal field is identically zero. This should now be more clear.

* I am confused by the discussion of the Gauge fixing condition $C_2=0$. It appears that for $C_2=0$ to be satisfied, it is necessary that Eq. 7 is satisfied, as this seems to be required in order that Eq. 11 satisfies Eq. 10. However, it would seem Eq. 17 does not in general satisfy Eq. 7. It only satisfies it when $\alpha=1$, for other values it does not seem to be normalized as required to satisfy Eq. 7. Since $C_2=0$ is a gauge fixing condition, this doesn't seem to actually be a problem for the main results, however the presentation seems to be written as if all choices of g considered lead to C_2 being zero.

With the new additions to the manuscript what was Eq. (7) is now Eq. (14) and what was Eq. (17) is now Eq. (26).

We have clarified the explanation of the reviewer's points by including additional text (in red) throughout the section. We want to emphasise that no results have been changed in any way and we detected no errors in the formalism. All changes simply amount to a more detailed explanation.

We now make clear the distinction between the longitudinal green's function g_L , which is fixed uniquely by Eq. (14), and the transverse part g_T , which Eq. (14) does not constrain in any way. The referee's concern seemed to be that g_T was eventually being specified (in what is now Eq. (26)) in such a way as to actually violate the equation that defines it, which it seems the referee was taking as Eq. (14). We think that this misunderstanding is because the notation we have used, although consistent, was not adequately explained. In actual fact upon recognition of the Helmholtz decomposition which gives $g = g_L + g_T$ it should be clear that there are no inconsistencies. Eq. (14) will always be satisfied by the full green's function g and the longitudinal part g_L independent of g_T . All that is required of g_T is that it is indeed transverse. In particular, g_T in what is now Eq. (26) is identically transverse independent of α , due to the presence of the transverse delta function.

The referee has we think noticed that, for $\alpha=1$ and for $x \neq x'$ the transverse part g_T in Eq. (26) does also happen to satisfy Eq. (14). This is because the divergence of g_T is identically zero and because the delta function on the RHS of Eq. (14) is also identically zero when $x \neq x'$. This fact is, however, entirely incidental. Independently of how we choose g_T , we have $C_2=0$ when Eqs. (14) and Eq. (19) hold. These equations are assumed to hold but they place no restrictions on g_T beyond that it is transverse. In other words we consider potentials of the form in Eq. (19) as a way of fixing $C_2=0$. This fixes the gauge only up to a choice of g_T , which is a c-number even at the quantum level. We subsequently parametrise the remaining gauge-freedom to choose g_T by means of the parameter α as specified in what is now Eq. (26). The only constraint that g_T needs to satisfy for the formalism to be consistent is transversality, and, g_T in what is now Eq. (26) is indeed transverse.

We hope that with the better explanation of the notation, combined with the inclusion of the appropriate properties of vector fields, it is clear that all equations are consistent. We have aimed to make all steps in the formalism clear.

* The discussion in supplemental section C is not clear. The first half of this section seems all to be predicated on the consequences of assuming that the TRK sum rule is satisfied for the two levels of the matter system. I do not understand why such an assumption would be considered. In most work I am aware of, the TRK sum rule is used on the total sum over all levels, and is then used as to provide an inequality for the transitions between the lowest two levels. i.e., for transitions from the ground state, one can know all terms are positive on the left hand side of Eq. 51, so each term individually must be less than Eq. 51. I am not aware of other works starting from the premise that in the two-level approximation the remaining term in the sum must be equal to the TRK sum.

We have added text so as to explain our motivation here better. We have also reorganised this section to make the motivation clearer. We attempt to explain better why we use the specific application of the TRK sum rule presented. We have tried to emphasise that this use of the sum rule is ad hoc, even spurious, and we do not necessarily advocate it, but we want to point out that this application is necessary *within* two-level models if one wants to elicit gauge-invariant energy shifts for even the low orders of perturbation theory. In this sense two-level truncations are already "inconsistent with basic quantum mechanics" (see Ref. 51). This drawback of two-level models is already well-known. We have noted also how the application of the TRK sum-rule described in this section is different to other applications in the ultrastrong light-matter physics literature.

For genuine atomic systems TRK sum rules are necessary in order to elicit gauge-invariance of level shifts e.g. the Lamb shift (see e.g. Refs. 38, 57). More precisely, to show the invariance of the Lamb shift at second order in perturbation theory when it is derived in e.g. the Coulomb and multipolar gauges one requires a way of relating the transition dipole moments to the mass of the dipole. This is precisely what the TRK provides. It is only with use of the TRK that the Coulomb and multipolar gauge expressions for the Lamb shift can be shown to be the same. Since a two-level truncation fundamentally alters the operator algebra, it is necessary in order to elicit the same gauge-invariance of energy levels derived within two-level models, that any sum rules relying on the original CCR algebra are used judiciously, i.e., are used in a way that accounts for the modification of the operator algebra.

The question we are asking in this section is not; what inequalities may we derive for the first two atomic levels based on the TRK sum-rule involving all dipole levels? We do recognise that, as the referee points out, this is often considered in the literature, but our motivation is different. Having shown in the main text that two-level models are non-equivalent between gauges, the question we are considering is; can gauge-invariance of level shifts be salvaged within two-level models in conventional regimes, despite the non-equivalence of the two-level models themselves? Equivalently, is there any systematic application of the TRK that can be used *within* the two-level truncation, i.e., within the modified algebra, which will give gauge-invariant level shifts in conventional regimes?

In this section of the SM we show that gauge-invariance of shifts found using non-equivalent two-level models can in fact be elicited up to second order in the coupling, provided we use the TRK sum rule in the way we present. We propose this ad hoc use of the TRK to favour the two-level truncations as much as possible. The point we

want to make here is that, even the most favourable of assessments of two-level truncations will at best restore consistency with gauge-invariance for level shifts at weak-coupling. No kind of consistency at strong coupling can be identified. If one is not favourable towards the truncation in the first place, i.e., one does not allow a somewhat ad hoc application of the TRK, then consistency cannot generally be identified at all (for weak or strong coupling).

We note finally that we have also expanded this section as an appropriate place to include a discussion of other possible ways in which a "two-level model" might be defined. A comparison of these different methods is left for further work. We have added remarks concerning this point in the conclusions section (in red) and in section III after Eq. (6).

* There is a typo on page 18, "differewnt"

Typo fixed.

REVIEWERS' COMMENTS:

Reviewer #1 (Remarks to the Author):

The authors have responded to all of my comments and made the necessary changes to the manuscript. I am satisfied with the revised version of the manuscript. I do not have any further comments.

Reviewer #2 (Remarks to the Author):

I have carefully read the revised version of the manuscript "Gauge ambiguities in QED: Jaynes-Cummings physics remains valid in the ultrastrong-coupling regime" by A. Strokes and A. Nazir. In the revised manuscript, the authors have successfully addressed all points raised in my previous report.

I recommend publication of this paper in Nature Communications in its present form.

Reviewer #3 (Remarks to the Author):

The revised manuscript by Stokes and Nazir has significantly improved the clarity of presentation, and has fully addressed the issues I raised previously. As a result the manuscript is much clearer, and I recommend it for publication. As noted previously, the manuscript makes a new contribution to the question of which gauge is best for writing approximate two-level models, and with the revisions now made, it also provides both a more convincing case of why the JC model is frequently best, as well as a clear understanding of what leads to this observation.

There are a few minor typos and points I noted on reading the revised manuscript:

* Page 1, typo: indespicable

* Page 1, typo: multiploar

* Page 2. As now explained in the appendix, in the main text $A=A_T$, and the subscript T is almost always suppressed in the main text. There is an exception in Eq. 1, which has an explicit T subscript, which presumably should be removed.

I think it would also help the reader to state more explicitly in the main text that A appearing in the main text is always the transverse part, and that the subscript is suppressed. (I acknowledge that it is introduced this way, but think it could be made even clearer.)

* Page 4, typo: multipoilar

* Page 4, top of second column. When introducing the eigenvalues ϵ_0 and ϵ_1 , it would help to state clearly these are eigenvalues of the material Hamiltonian, as this is not entirely clear.

* Page 4, Refs. 41,44-47 are about literature on the superradiant transition and no go theorem. In this list, it would be appropriate to include Ref. 40 as well (as this, more than Ref. 41 is about this transition). I think it would also be appropriate to include: [Vukics, Griesser, and Domokos, Phys. Rev. Lett. 112, 073601 (2014)] as being other recent relevant work on this question.

* Page 7, typo in caption "inest" for "inset"

- * Page 8, typo: predicitng
- * Page 9, typo, repeated full stop after Eq. 13
- * Page 10, I think the sentence "We consider bound charges..." really starts a new paragraph, as the subject changes from the properties of g to the effective Lagrangian
- * Page 10, after Eq. 17, typo "effect" should be "affect"
- * Page 10, I am unsure of using the words "gauge fixing condition" for a condition that restricts but does not entirely fix the gauge. i.e., as discussed below, within the constraint $C_2=0$, all the α gauges are still available. I wonder if this could be better phrased.
- * Page 10, typo "C_!" for "C_1"
- * Page 16, after Eq. 64, this discussion is rather unclear. Specifically, it is unclear what is meant by $R_{0\alpha}(y)$. Elsewhere y is defined as a list of the gauge dependent canonical coordinates, but an expression is given in this paragraph which for $\alpha=0$ seems to suggest $U_0 = R_{00}(P^0 y p^0) = 1$. This doesn't seem consistent, as it's unclear how the list of coordinates becomes the identity.
- * Page 16/17, section VI.D. This section presents a perturbative approach as if it were invented in Ref. 54. However, (as in fact stated in Ref. 54), this perturbative approach is the standard Schrieffer-Wolff perturbation theory (sometimes also known as Van Vleck perturbation theory). It would help the reader to point this out.
- * Page 20, section reference to VI.E presumably means VI.D

Comments of Referee 3

There are a few minor typos and points I noted on reading the revised manuscript:

- * Page 1, typo: indespicable
- * Page 1, typo: multiploar
- * Page 2. As now explained in the appendix, in the main text $A=A_T$, and the subscript T is almost always suppressed in the main text. There is an exception in Eq. 1, which has an explicit T subscript, which presumably should be removed.
- * Page 4, typo: multipoilar
- * Page 7, typo in caption "inest" for "inset"
- * Page 8, typo: predicitng
- * Page 9, typo, repeated full stop after Eq. 13
- * Page 10, I think the sentence "We consider bound charges..." really starts a new paragraph, as the subject changes from the properties of g to the effective Lagrangian
- * Page 10, after Eq. 17, typo "effect" should be "affect"
- * Page 10, typo "C_!" for "C_1"
- * Page 20, section reference to VI.E presumably means VI.D

All of the above typos/errors have been rectified.

I think it would also help the reader to state more explicitly in the main text that A (referring to the vector potential) appearing in the main text is always the transverse part, and that the subscript is suppressed. (I acknowledge that it is introduced this way, but think it could be made even clearer.)

The notation has now been explained more clearly where it is introduced

* Page 16/17, section VI.D. This section presents a perturbative approach as if it were invented in Ref. 54. However, (as in fact stated in Ref. 54), this perturbative approach is the standard Schrieffer-Wolff perturbation theory (sometimes also known as Van Vleck perturbation theory). It would help the reader to point this out.

We have now pointed out the origin of the approach where it is introduced.

* Page 16, after Eq. 64, this discussion is rather unclear. Specifically, it is unclear

what is meant by $R_{0\alpha}(y)$. Elsewhere y is defined as a list of the gauge dependent canonical coordinates, but an expression is given in this paragraph which for $\alpha=0$ seems to suggest $U_0 = R_{00}(P^0 y p^0) = 1$. This doesn't seem consistent, as it's unclear how the list of coordinates becomes the identity.

We have now clarified this point. The notation $R(y)$ indicates functional dependence of R on y . It does not denote R acting on y . The equality $U_0 = R_{00}(P^0 y p^0) = 1$ follows because $R_{00}=1$ identically independent of its argument. This follows immediately from the definition of R , recalling that $R_{\alpha\alpha}$ is defined as the exponent of a term depending on $\alpha-\alpha$.

* Page 4, top of second column. When introducing the eigenvalues ϵ_0 and ϵ_1 , it would help to state clearly these are eigenvalues of the material Hamiltonian, as this is not entirely clear.

This has now been clarified

* Page 4, Refs. 41,44-47 are about literature on the superradiant transition and no go theorem. In this list, it would be appropriate to include Ref. 40 as well (as this, more than Ref. 41 is about this transition). I think it would also be appropriate to include: [Vukics, Griesser, and Domokos, Phys. Rev. Lett. 112, 073601 (2014)] as being other recent relevant work on this question.

This reference has been added

* Page 10, I am unsure of using the words "gauge fixing condition" for a condition that restricts but does not entirely fix the gauge. i.e., as discussed below, within the constraint $C_2=0$, all the α gauges are still available. I wonder if this could be better phrased.

We have reworded this so we now refer to the constraint as a constraint on the form of $\{A\}$.

OTHER CHANGES TO MAIN TEXT

We have moved parts of the main text to the methods section so as to shorten the main text to be within the 5000 word limit. No material has been removed.

We have streamlined/reworded certain paragraphs to shorten the text (e.g. the paragraph about super-radiant phase transition in the Dicke model on page 4). The explanatory content remains the same.

Graphs have been remade to be consistent with requirements.

Abstract has been shortened and title has changed to be consistent with requirements.

Section numbering and labelling changed to be consistent with requirements.

Introduction/results reorganised to be consistent with requirements.

Discussions/conclusions reorganised to be consistent with requirements

CHANGES TO SUPPLEMENTARY INFORMATION

Re-formatted to be consistent with requirements

We removed Fig. 2(b) as it is not needed and had not been referred to in the text. What was Fig. 2(a) is now just Fig. 2. We have added an additional curve to Fig. 2 which now includes both renormalised and non-renormalised ratios to allow better comparison. Previously only the renormalised ratio was shown.

Eqs. (78) and (79) were missing powers 2 on certain bracketed parts. This has been rectified.

Supplementary note 6 about photon number has been expanded to give more explanation of how plots are obtained. We found a formatting error in the final plot 14 whereby the wrong values were printed for the upper bound on the y-axis. This has now been amended. For completeness, we have added some additional curves to Fig. 14. We now show the JC-gauge two-level model prediction in the excited state and we show predictions for both possible definitions of the flux-gauge projection of the JC-gauge photon number operator.